# tRNA as an assembly chaperone for a macromolecular transcription-processing complex

Julia Bartuli[1,9], Stefan Jungwirth[1], Manisha Dixit[1,8], Takumi Okuda [2],
Johannes Patrick Zimmermann [3], Matthias Erlacher [4], Tao Pan [5],
Asisa Volz[6], Alexander Hüttenhofer[4], Bettina Warscheid [3],
Claudia Höbartner [2], Clemens Grimm [1,9] ✉ & Utz Fischer [1,7] ✉

Transfer RNAs (tRNAs) are widely recognized for their role in translation. Here, we describe a previously unidentified function of tRNA as an assembly chaperone. During poxviral infection, tRNA$^{Gln/Arg}$ lacking the anticodon mcm$^5$s$^2$U34 modification is specifically sequestered from the cellular tRNA pool to promote formation of a multisubunit poxviral RNA polymerase complex (vRNAP). Cryo-electron microscopy analysis of assembly intermediates illustrates how tRNA$^{Gln/Arg}$ orchestrates the recruitment of transcription and mRNA processing factors to vRNAP where it controls the transition to the preinitiation complex. This is achieved by an induced fit mechanism that internalizes anticodon base G36 into the anticodon stem, creating a noncanonical tRNA structure and selecting a defined tRNA modification pattern. The role of tRNA as an assembly chaperone extends to the pathogenic Mpox virus, which features a similar vRNAP.

Transfer RNAs (tRNAs) are best known for their role in decoding mRNA codons and translating them into proteins[1]. However, certain tRNAs are also involved in a variety of noncanonical functions including nutrient sensing[2], splicing[3], transcription[4], apoptosis[5] and scaffolding[6], as reviewed by Su et al.[7]. In these contexts, tRNAs or their fragments can act as antisense decoys, protein modulators, primers or sensors. Recently, human tRNA$^{Gln}$ and, to a lesser extent, tRNA$^{Arg}$ have been identified in a cellular context that is not consistent with any of the established functions of tRNAs. Specifically, these tRNAs (termed tRNA$^{Gln/Arg}$ throughout this manuscript) were found to be a stoichiometric component of a macromolecular RNA polymerase complex, known as complete vRNAP, which forms in cells upon infection with the

prototypic poxvirus vaccinia[8]. This virus belongs to the diverse group of nucleocytoplasmic large DNA viruses, comprising double-stranded DNA viruses that express their genome within the cytoplasm of their host using their own gene expression machinery[9,10]. The megadalton complete vRNAP unit integrates the poxviral core RNA polymerase (core vRNAP), composed of eight Rpo subunits, with early transcription factors Rap94, VETF-s, VETF-l, NPH-I and E11, the capping enzyme dimer D1/D12 and host tRNA$^{Gln/Arg}$.

We showed previously that recruited tRNA$^{Gln/Arg}$ is uncharged and tethers associated factors to vRNAP through interactions with Rap94, NPH-I and VETF-l (ref. 8). Consistent with its biochemical composition, complete vRNAP acts as an autonomous early transcription unit

[1]Department of Biochemistry 1, Theodor Boveri-Institute, University of Würzburg, Würzburg, Germany. [2]Institute of Organic Chemistry, University of Würzburg, Würzburg, Germany. [3]Department of Biochemistry 2, Theodor Boveri-Institute, University of Würzburg, Würzburg, Germany. [4]Institute of Genomics and RNomics, Biocenter, Medical University of Innsbruck, Innsbruck, Austria. [5]Department of Biochemistry and Molecular Biology, University of Chicago, Chicago, IL, USA. [6]University of Veterinary Medicine Hannover (TiHo), Institute of Virology, Hannover, Germany. [7]Helmholtz Institute for RNA-based Infection Research (HIRI), Helmholtz Centre for Infection Research (HZI), Würzburg, Germany. [8]Present address: National Heart Lung and Blood Institute, National Institutes of Health, Bethesda, MD, USA. [9]These authors contributed equally: Julia Bartuli, Clemens Grimm. ✉e-mail: clemens.grimm@uni-wuerzburg.de; utz.fischer@uni-wuerzburg.de

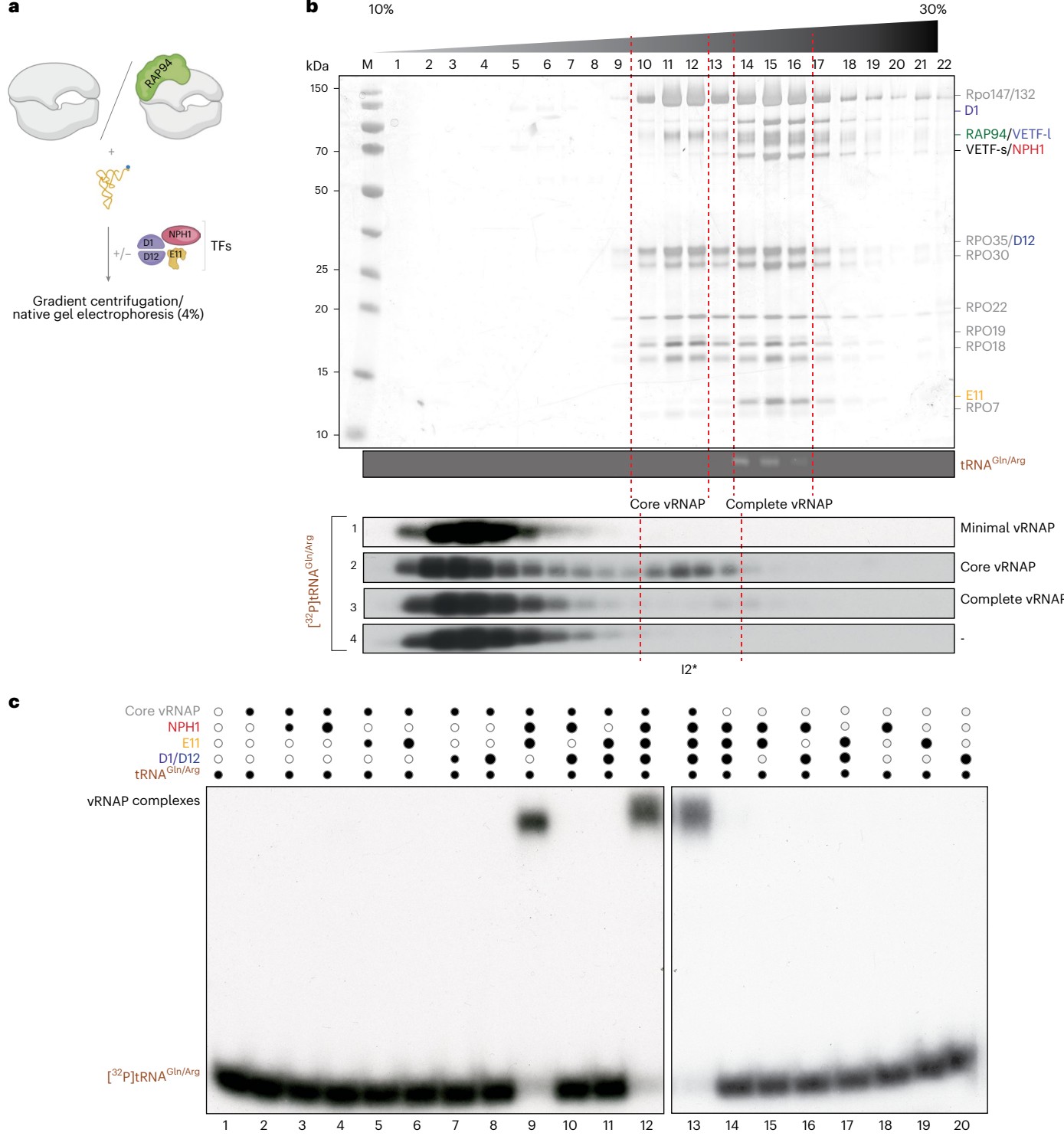

**Fig. 1 | Reconstitution of vRNAP intermediates. a**, Schematic representation of the biochemical reconstitution strategy of vRNAP assembly intermediates. Minimal and core vRNAPs were isolated from infected cells by anti-FLAG affinity chromatography; factors D1/D12, E11 and NPH-I were expressed in *E. coli* and purified as described in the Methods. tRNA^Gln/Arg^ was obtained from purified complete vRNAP as described in the Methods. **b**, Top, silver-stained protein gel of fractions from a sucrose density gradient centrifugation of isolated vRNAP complexes. tRNA^Gln/Arg^ from purified vRNAP was visualized by ethidium bromide staining. Fractions 10–12, minimal and core vRNAP sediment; fractions 14–16,

complete vRNAP sediments. Bottom, [^32P]tRNA^Gln/Arg^ was incubated with the indicated vRNAP complexes, separated by gradient centrifugation and analyzed by autoradiography. The experiment was performed in triplicate. **c**, Gel mobility shift assays of core vRNAP and recombinant factors in the presence of [^32P]tRNA^Gln/Arg^. Black dots, added components; white dots, omitted components. The dot size corresponds to the amounts of recombinant factors added. Complex formation was visualized by autoradiography. The experiment was performed four times.

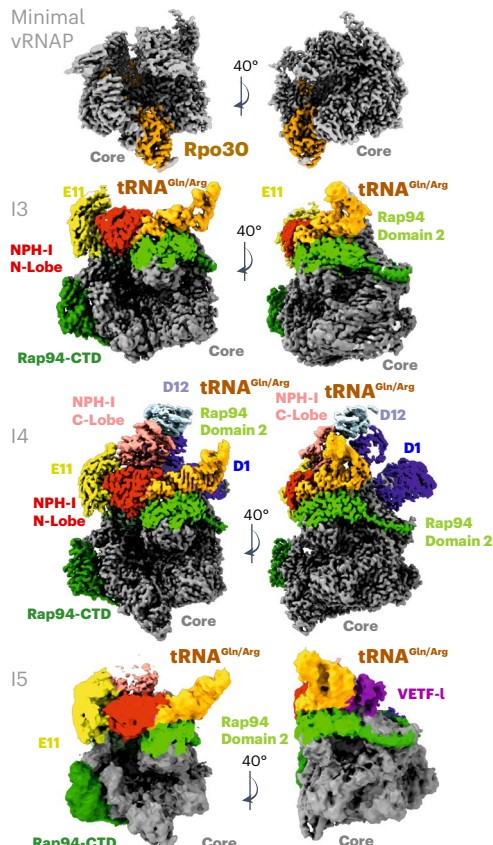

**Fig. 2 | Experimental key structures of the early PIC tRNA-chaperoned assembly cycle.** Isosurface representations of cryo-EM structures are shown and crucial structural elements are labeled. The intermediates are labeled I3–I5 according to their occurrence in the presumed assembly cycle. The color code of the individual factors is identical to that of Figs. 1, 3 and 4. Each structure is shown in a standard view that corresponds to that used in Fig. 3 and an additional view with 40° rotation. Rpo30 is highlighted in ochre only for minimal vRNAP. Note that, in general, all bound factors occupy corresponding positions; therefore, the visualized structures document an incremental buildup, eventually yielding to complete vRNAP in the absence of major conformational reorganizations (full assembly cycle in Fig. 3).

capable of generating m⁷G-capped transcripts[8,11]. Notably, tRNA$^{Gln/Arg}$, although part of this complex, is not directly involved in transcription and is absent from all DNA-bound transcription complexes identified to date. We, therefore, hypothesized that these tRNAs do not function as transcription factors but rather as chaperones that control the association of vRNAP with adjunct factors required for early transcription. Here, we combined a biochemical reconstitution system with structural analysis by cryo-electron microscopy (cryo-EM) to investigate the assembly pathway of complete vRNAP. Our study uncovers an unknown function of a specific tRNA as an assembly chaperone and reveals a unique induced fit mechanism that involves structural rearrangement of the tRNA, enabling complex formation.

## Results

### Coordinated assembly of Rap94, NPH-I and E11 with vRNAP by tRNA$^{Gln/Arg}$

To test whether tRNA$^{Gln/Arg}$ acts as an assembly chaperone rather than a scaffold, we dissected the steps that lead to the formation of complete vRNAP. We isolated a minimal version of vRNAP, consisting of the eight Rpo subunits, only as a presumed starting point of assembly. This unit, along with recombinant transcription factors (Extended Data Fig. 1a) and tRNA$^{Gln/Arg}$ isolated from complete vRNAP, was then used to study

the formation of complete vRNAP in vitro (Fig. 1a). Cosedimentation and gel mobility shift experiments showed that [³²P]tRNA$^{Gln/Arg}$ alone failed to bind minimal vRNAP (Fig. 1b, bottom, gradient profiles 1–4) or any of the individual transcription factors (Fig. 1c, lanes 14–20). However, tRNA$^{Gln/Arg}$ bound weakly to core vRNAP (minimal vRNAP associated with Rap94), forming the first detectable, albeit unstable, tRNA-associated vRNAP complex (I2*; Fig. 1b, bottom). Transition to specific and stable tRNA$^{Gln/Arg}$ binding occurred only in the presence of Rap94, NPH-I and E11, as demonstrated by band shift assays (Fig. 1c, lane 9), whereas omission of any of the adjunct factors prevented complex formation (Fig. 1c, lanes 2–8,10 and 11). Of note, none of these factors bound stably to each other (Extended Data Fig. 1b) or to core vRNAP in the absence of tRNA$^{Gln/Arg}$ (Extended Data Fig. 1c), illustrating its crucial role in assembly. However, weak background binding activity can be attributed to existent invariable contact surfaces involving these three components, as documented in the structure of complete vRNAP. Once tRNA$^{Gln/Arg}$ was stably bound (Fig. 1c, lane 9), D1/D12 also readily associated with vRNAP (Fig. 1c, lanes 12 and 13) whereas the VETF-s/l heterodimer was not required for these assembly steps, as discussed below. The fully assembled complete vRNAP exhibited remarkable stability, as evidenced by its inability to bind or exchange [³²P]tRNA$^{Gln/Arg}$ (Fig. 1b, bottom, gradient profiles 3 and 4).

### Structural insights into the vRNAP assembly pathway

The simultaneous requirement of E11, NPH-I, Rap94 and tRNA$^{Gln/Arg}$ for the formation of complete vRNAP suggested a complex binding mechanism, potentially involving conformational changes among interacting partners. Using our biochemical reconstitution system, we leveraged cryo-EM to decipher the mechanism of assembly by visualizing critical intermediates. Atomic structures of the intermediates (labeled I1–I5) were then used to propose an assembly cycle that configures complete vRNAP and enables its subsequent conversion into the preinitiation complex (PIC), ready to initiate transcription at an early promoter (Figs. 2 and 3).

We first determined the structure of minimal vRNAP (Fig. 2, Extended Data Fig. 2 and Table 1). A comparison with the already known

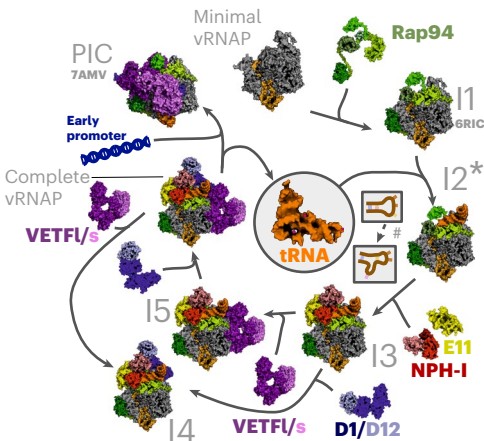

**Fig. 3 | The tRNA-chaperoned assembly cycle.** A model of the tRNA-chaperoned assembly cycle was derived from biochemical and structural data of this study and is illustrated by solved and available atomic structures, the latter of which are labeled with their PDB codes. tRNA refers to tRNA$^{Gln/Arg}$. Assembly intermediates, shown as accessible molecular surface, are labeled according to their appearance during the assembly cycle: I1, core vRNAP (PDB 6RIC); I2*, metastable core vRNAP–tRNA (no experimental structure available); I3, core vRNAP–tRNA$^{Gln/Arg}$–E11–NPH-I; I4, complete vRNAP–VETF-s/l; I5, complete vRNAP lacking the capping enzyme (complete vRNAP–D1/D12). The starting point is minimal vRNAP and the endpoint is PIC on an early promoter (PDB 7AMV). *Transient intermediate that can be detected only by biochemical means (Fig. 1b, bottom, gradient profile 2). #Conformational change in the tRNA assembly chaperone.

**Table 1 | Data collection, cryo-EM and model refinement statistics**

| | I5 (complete vRNAP without D1/D12) EMD-16476, PDB 8C8H | Complete vRNAP EMD-19442, PDB 8RQK | I3 EMD-50639, PDB 9FPY | I4 EMD-50644, PDB 9FQ6 | Minimal vRNAP EMD-50033, PDB 9EX9 |
|---|---|---|---|---|---|
| **Data collection and processing** | | | | | |
| Magnification | 75,000 | 75,000 | 75,000 | 130,000 | 130,000 |
| Voltage (kV) | 300 | 300 | 300 | 300 | 300 |
| Electron exposure (e⁻ per Å²) | 70 | 78 | 40 | 40 | 40 |
| Defocus range (μm) | −1.0 to −2.2 | −1.0 to −2.2 | −1.0 to −2.0 | −1.0 to −2.0 | −1.0 to −2.0 |
| Pixel size (Å) | 1.0635 | 1.0635 | 0.964 | 0.964 | 0.964 |
| Symmetry imposed | $C_1$ | $C_1$ | $C_1$ | $C_1$ | $C_1$ |
| Initial particle images (no.) | 188,000 | 1,753,500 | 4,738,448 | 4,738,448 | 1,016,117 |
| Final particle images (no.) | 21,338 | 934,606 | 148,102 | 171,057 | 404,153 |
| Map resolution (Å) | 3.9 | 2.65 | 2.5 | 2.5 | 2.5 |
| FSC threshold | 0.143 | 0.143 | 0.143 | 0.143 | |
| Map resolution range (Å) | 3.4–21.2 | 2.35–13.8 | 2.2–40.8 | 2.2–40.6 | 2.13–40.1 |
| **Refinement** | | | | | |
| Initial model used (PDB code) | 6RFL | 6RFL | 6RFL | 6RFL | 6RFL |
| Model resolution (Å) | 3.8 | 2.6 | 2.6 | 2.6 | 2.5 |
| FSC threshold | 0.143 | 0.143 | 0.143 | 0.143 | 0.143 |
| Model resolution range | 3.8–20 | 2.6–20 | 2.6–20 | 2.6–20 | 2.5–20 |
| Map sharpening $B$ factor (Å²) | −99 | −85 | −15 | −15 | −65 |
| **Model composition** | | | | | |
| Nonhydrogen atoms | 41,296 | 52,462 | 41,025 | 51,728 | 26,970 |
| Protein residues | 4,915 | 6,266 | 4,858 | 6,178 | 3,345 |
| Nucleic acid residues | 63 | 72 | 72 | 72 | 0 |
| Ions | Zn, 4; Mg,1 | Zn, 4; Mg, 4 | Zn, 4; Mg, 1 | Zn, 4; Mg, 1 | Zn, 4; Mg, 1 |
| Waters | 0 | 9 | 0 | 0 | 0 |
| **$B$ factors (Å²)** | | | | | |
| Protein | 148.2 | 158.3 | 112.0 | 73.1 | 92.7 |
| Nucleic acid | 149.7 | 189.1 | 95.9 | 120.0 | - |
| **Root-mean-square deviations** | | | | | |
| Bond lengths (Å) | 0.002 | 0.005 | 0.003 | 0.003 | 0.005 |
| Bond angles (°) | 0.56 | 0.50 | 0.52 | 0.52 | 0.51 |
| **Validation** | | | | | |
| MolProbity score | 2.3 | 1.8 | 1.6 | 1.6 | 1.3 |
| Clashscore | 7.3 | 4.8 | 3.4 | 4.5 | 2.7 |
| Poor rotamers (%) | 4.2 | 2.32 | 0.99 | 1.00 | 0.85 |
| **Ramachandran plot (%)** | | | | | |
| Favored | 93.4 | 94.8 | 94.8 | 94.3 | 96.0 |
| Allowed | 6.6 | 5.2 | 5.2 | 5.7 | 4.0 |
| Disallowed | 0.0 | 0.0 | 0.0 | 0.0 | 0.0 |

FSC, Fourier shell correlation.

structure of core vRNAP (assembly intermediate I1)[8] (Fig. 3) illustrates how the multidomain protein Rap94 wraps around minimal vRNAP during the first step of the assembly process. In doing so, Rap94 adopts a conformation that remains stable throughout the whole assembly cycle (Extended Data Fig. 3). In agreement with our biochemical experiments (Fig. 1b), binding of tRNA^Gln/Arg to core vRNAP (assembly intermediate I2*) turned out to be too transient for structural analysis. However, the stable assembly intermediate I3 was accessible for cryo-EM reconstruction, revealing how E11 connects with NPH-I and

facilitates tRNA binding to core vRNAP (Fig. 2 and Extended Data Fig. 4). tRNA^Gln/Arg is anchored through extensive interactions of its D-arm and T-arm and the anticodon stem with Rap94. Upon NPH-I binding, its N-lobe squeezes the tRNA^Gln/Arg anticodon stem loop (ASL) into a strained conformation. Notably, the interface of Rap94 with tRNA^Gln/Arg does not depend on specific base contacts; instead, several bases in the ASL are specifically recognized by the NPH-I C-lobe, as discussed below.

Full stability of the vRNAP complex is achieved by independently recruiting the VETF-s/l and D1/D12 heterodimers, forming either

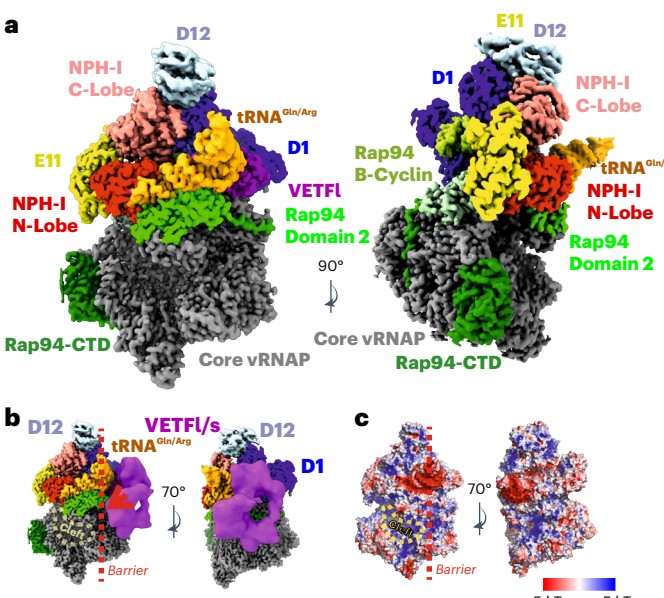

**Fig. 4 | High-resolution reconstruction of complete vRNAP. a**, An isosurface representation of the cryo-EM density with crucial structural elements labeled. The color code of the individual factors is identical to that of Figs. 2 and 3. **b**, A similar isosurface representation to **a**, overlaid with a 20-Å lowpass-filtered density depicted as a magenta isosurface. This density can be attributed to mobile VETF. **c**, Electrostatic potential mapped to the complete vRNAP solvent-accessible surface. The orientation of both views is identical to those depicted in **b**. The lowpass-filtered density overlay shown in **b** is not presented here.

complete vRNAP–VETF (I4; Fig. 2, Extended Data Fig. 4 and Supplementary Table 1) or complete vRNAP–D1/D12 (I5; Fig. 2, Extended Data Fig. 5 and Supplementary Table 1), depending on which factor binds first. It is noteworthy that I5 was found in native vRNAP preparations, whereas I4 was only observed during reconstitution studies in vitro. It stands to reason that both intermediates enable alternative pathways leading to the assembly of complete vRNAP (Figs. 3 and 4). Notably, complete vRNAP is characterized by exceptional stability because of a belt-like structure formed by E11, NPH-I, tRNA[Gln/Arg], VETF-s/l and D1/D12 on the surface of core vRNAP (Fig. 4).

As shown previously[12], complete vRNAP can bind to early viral promoters and reconfigures under the elimination of E11, NPH-I, D1/D12 and tRNA[Gln/Arg] into the PIC. The transition of complete vRNAP to the PIC is tightly regulated by tRNA[Gln/Arg], which adopts a pose in complete vRNAP that blocks the access of VETF-s/l to the cleft in the absence of a cognate promoter (Fig. 4b). The presence of a cognate promoter, by contrast, enables the displacement of tRNA[Gln/Arg] and PIC formation as the endpoint of the assembly pathway. tRNA[Gln/Arg], hence, orchestrates an assembly pathway without being a component of complete vRNAP, the final product (Fig. 3 and Supplementary Video 1).

### tRNA binding involves an unusual, strained conformation of the ASL

To explore the details of the tRNA–protein interactions enabling the assembly process, we generated a high-resolution cryo-EM dataset of complete vRNAP and rerefined our previous structural model (Figs. 4a and 5a, Extended Data Fig. 6 and Table 1). Upon superimposing the structure of tRNA[Gln] as part of complete vRNAP with other known tRNA structures from the Protein Data Bank (PDB), we identified notable deviations in the base-pairing pattern of the anticodon stem. These differences were associated with a strained backbone conformation and the coordination of a magnesium ion (Fig. 5a–c). Similar rearrangements are possible for tRNA[Arg] (Extended Data Fig. 7), as discussed below.

Because no other experimental structure for human tRNA[Gln](UUG) existed in the database, we modeled its ground state starting from the available structure of bacterial tRNA[Gln][13] and performed energy minimization. In the ground state, the three anticodon bases (U34, U35 and G36) are turned outward and U33 pairs with A37, forming the most distal base pair of the anticodon stem (Fig. 5d–f). In comparison, the most notable feature of the strained ASL conformation is the inward rotation of the anticodon base G36, which forms a hydrogen bond with U33, causing A37 to bulge outward (Fig. 5a–c) into a pocket located in the N-lobe of NPH-I (Fig. 5g). Therefore, the strained ASL conformation in complete vRNAP sharply contrasts with the relaxed ASL conformation (ground state) of free tRNA (Fig. 5b–e).

Lastly, we examined the RNA–protein interactions that might induce this strained conformation. In tRNA[Gln](UUG), the anticodon sequence comprises the bases U34, U35 and G36 (asterisks in Fig. 5a–g). The anticodon and the distal region of the anticodon arm are closely associated with the N-lobe of NPH-I, whose surface acts as a template, shaping the unusual stem-loop conformation. Interestingly, there is no specific interaction between the protein and the nucleic acid bases in most tRNA regions, except for a selective pocket accommodating U35, which seems poorly suited for any base other than a pyrimidine (Fig. 5g). This observation suggests that the selectivity for tRNA[Gln] (and tRNA[Arg], as discussed below) cannot be attributed to protein–tRNA interactions alone.

### Anticodon readout through internal base pairing within the anticodon stem

To understand the strict selectivity for tRNA[Gln/Arg] for the assembly of complete vRNAP, we next investigated the internal interactions and sequence constraints within the bound tRNA. Three specific conditions must be met for this interaction to occur. First, a noncanonical base pair must form between bases 33 and 36 at the distal position of the anticodon stem. Here, the imino group (N1) of G36 acts as a hydrogen-bond donor, binding to the carbonyl oxygen atom at C2 of U33 (limitation 1; Fig. 5b,c, yellow arrow, Supplementary Video 2, Extended Data Fig. 8, and Supplementary Table 1). This interaction requires a cytosine as the first base of the corresponding codon, as only a guanine at the anticodon position provides the necessary hydrogen-bond donor (Fig. 5h, yellow background). Second, the anticodon base at position 35 fits into a hydrophobic pocket on NPH-I that can only accommodate a pyrimidine (limitation 2; Fig. 5g, green arrow), necessitating a purine in the codon sequence (Fig. 5h, green background). Lastly, a noncanonical base pair must form between C32 and A38 in the anticodon stem. This pairing is compatible with five ASL sequences found in tRNA[Gln](UUG/CAG) and tRNA[Arg](UCG/CCG/ACG) but not with those found in tRNA[His], tRNA[Pro] or tRNA[Leu] (condition 3; Fig. 5h,i, and red arrow and background). Notably, tRNA[Arg](CCG) isoforms 1-1, 1-2 and 1-3 deviate in that they feature a canonical U32–A38 base pair (Fig. 5i). These constraints ensure that only tRNA[Gln] and tRNA[Arg] can act as assembly factors for complete vRNAP. Accordingly, these tRNA species are exclusively selected upon poxviral infection, determined by quantitative tRNA sequencing of isolated complete vRNAP, as discussed below. We conclude that complete vRNAP assembly occurs through a novel tRNA recognition mechanism, where anticodon readout is determined by internal base pairing.

### tRNA modification as a driver for complex formation

tRNAs belong to the most extensively modified noncoding RNA species[14,15] and it was a possibility that specific modifications of tRNA[Gln/Arg] influence its role in assembling the complete vRNAP complex. Supporting this hypothesis, we found that unmodified tRNA[Gln] produced by transcription in vitro was less effective in promoting vRNAP assembly compared to tRNA[Gln/Arg] extracted from complete vRNAP (Fig. 6a and Extended Data Fig. 8).

Using liquid chromatography–tandem mass spectrometry (LC–MS)[16] we determined the modification spectrum of vRNAP-associated

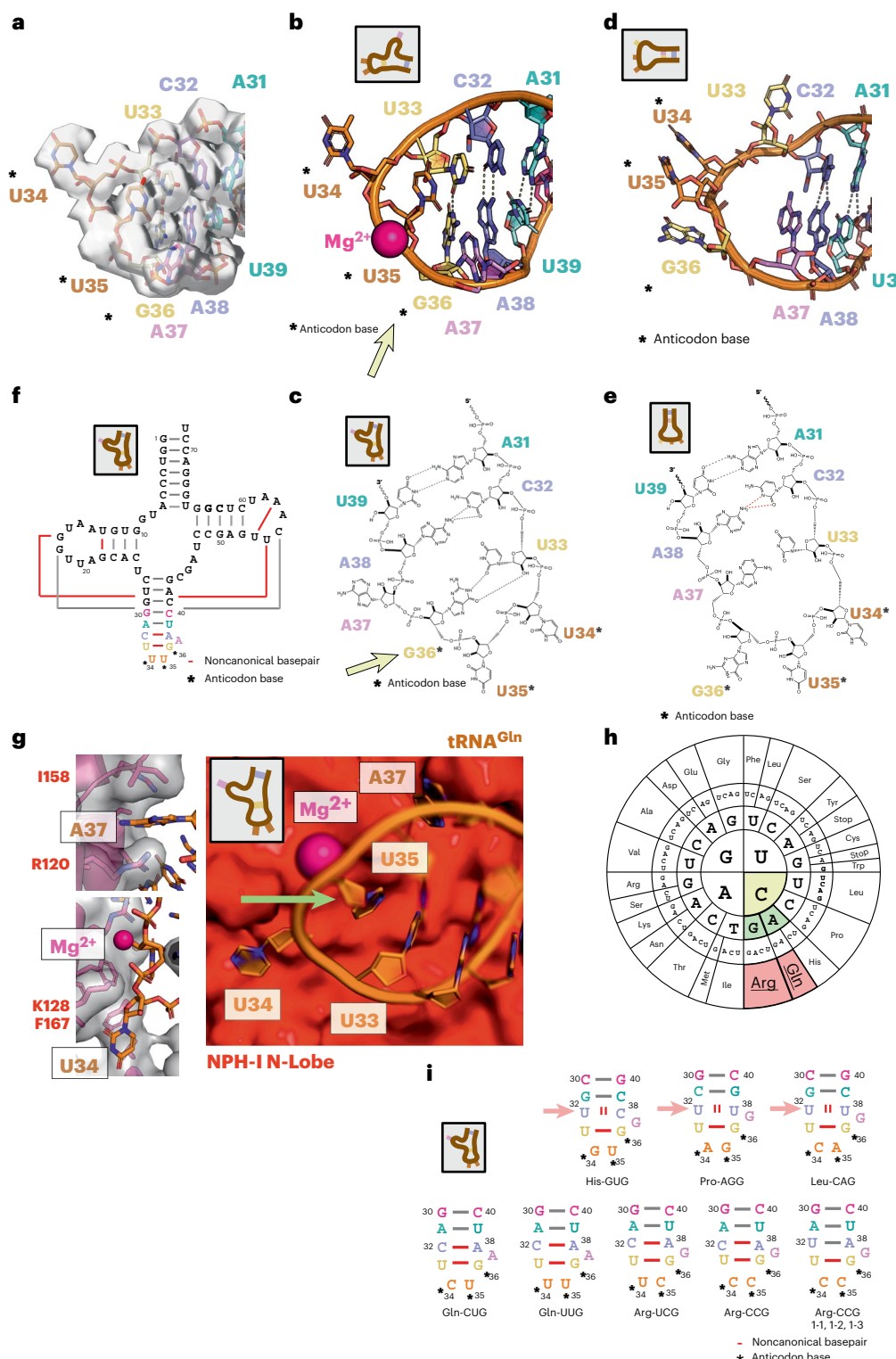

**Fig. 5 | Conformation and interaction of tRNA^Gln (UUG) in complete vRNAP.**
**a**, ASL structure from the cryo-EM model of complete vRNAP (strained state) depicted as a stick model, overlaid with the cryo-EM density shown as a transparent isosurface. **b**, ASL structure from the cryo-EM model of complete vRNAP depicted as cartoon-and-stick model. Please note that the relaxed state (free tRNA) is a manually created model because no experimental structure of this state is available (left, bound strained state; right, free relaxed state). **c,d**, Sketch of the ASL structure in the strained state (**c**) and in the relaxed state (**d**; free tRNA). Please note that this is a manually created model because no experimental structure of this state is available. **e**, Sketch of the ASL structure in the relaxed state. **f**, Scheme of the tRNA^Gln (UUG) base pairing in the bound strained state. **g**, Right, ASL region in cartoon-and-stick depiction with the adjacent NPH-I solvent-accessible surface (red) as observed in the model of complete vRNAP (strained state). Left, details of the NPH-I–tRNA interaction in the region of A37 and U34. **h**, Codon 'sun' with recognition-relevant bases marked in color. **i**, ASL base-pairing scheme for tRNA^Gln (UUG) as observed in complete vRNAP in the strained state (bottom left). Sequences of several other tRNA species are threaded into the scheme for the other depictions. Unfavorable base pairing is indicated by two red bars; canonical base pairs are shown in gray and nonconical base pairs are shown in red. If present, pictograms in insets symbolize the relaxed (unbound) or strained (bound) states (Supplementary Video 2). Please note that there is no high-confidence tRNA^Arg (GCG) sequence available in the database.

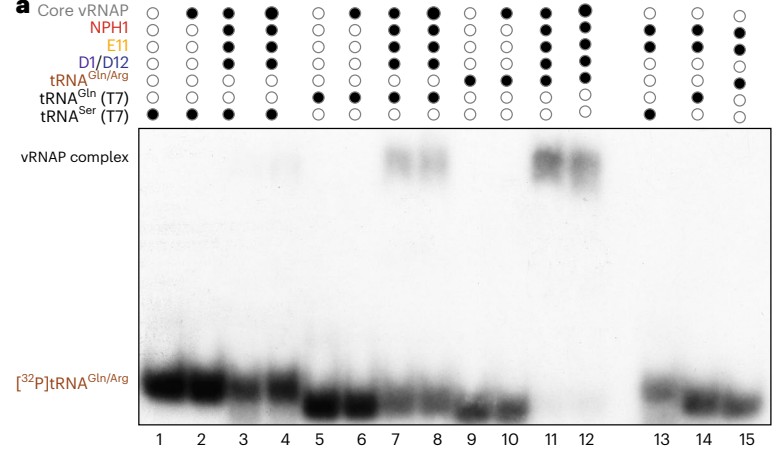

**Fig. 6 | Identification of modifications of tRNAGln/Arg and their impact for vRNAP formation. a**, Gel mobility shift assay of nonmodified (T7-transcribed) tRNASer (lanes 1–4, 13) and tRNAGln (lanes 5–8, 14) or modified tRNAGln/Arg (extracted from complete vRNAP; lanes 9–12, 15) in the presence of indicated adjunct factors E11, NPH-I and D1/D12 and core vRNAP. **b**, LC–MS analysis of modifications of tRNAGln/Arg isolated from complete vRNAP. **c**, LC–MS analysis of mcm5s2U and cm5s2U modifications in total tRNA isolated from infected cells. **d**, LC–MS analysis mcm5s2U and cm5s2U modifications in total tRNA isolated from uninfected cells. **e**, *K. lactis* γ-toxin digestion of tRNAs. The indicated

[32P]tRNAs (lines 1–14) were incubated with increasing concentrations γ-toxin (50 nM, 100 nM, 250 nM, 500 nM, 1 μM and 5 μM) for 1 h at 25 °C. RNA was subsequently phenol-extracted, resolved by denaturing RNA gel electrophoresis and visualized by radioautography. In lanes 15–18, total tRNA from infected cells was digested with increasing concentrations of γ-toxin (500 nM, 1 μM and 5 μM), followed by TRIzol extraction. tRNAs were separated on a 12% denaturing gel, transferred to a nylon membrane and hybridized with a 5′-end-labeled DNA probe complementary to the 3′-end of the tRNAGln (UUG) anticodon region. FL, full-length tRNAs; Dig.-[32P]tRNAs, digested tRNA fragments.

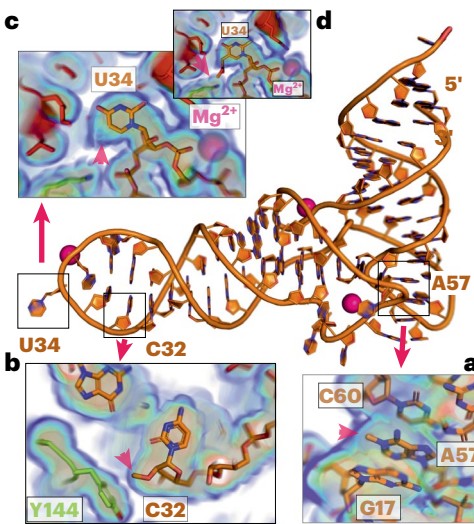

**Fig. 7 | Identification of modifications of tRNA[Gln] in the cryo-EM density of complete vRNAP. a**, Density and model at tRNA base A57. **b**, Density and model at tRNA base C32. **c**, Density and model at tRNA base U34. The red arrow indicates the expected position of the absent mcm⁵s²U3 modification. **d**, Model of the mcm⁵s²U34 modification. The model of mcm⁵s²U at position 34 of the complete vRNAP model produces a clash or unfavorable interaction with surrounding residues from Rap94 and NPH-I (red arrow).

tRNA[Gln/Arg] and found the common modifications m¹A, m⁵C, Cm, m¹G, Gm and pseudouridine (Fig. 6b). We then evaluated whether the modified nucleotides, positioned as predicted, were consistent with their chemical environment in the complete vRNAP model and the surrounding cryo-EM density. Two positions met these criteria. For nucleotide 57 in the tRNA elbow, m¹A fit the observed cryo-EM density better than its unmodified counterpart (Fig. 7a). In the anticodon stem at C32, Cm aligned well with the density and occupied a hydrophobic pocket partially formed by Y144 of Rap94 (Fig. 7b). The mcm⁵s²U modification at the wobble position U34 is a prominent feature of tRNA[Gln] (ref. 17). Interestingly, this modification was missing in tRNA extracted from complete vRNAP (Fig. 6b). However, it was present in tRNAs from both infected (Fig. 6c) and noninfected cells (Fig. 6d), although it was less prevalent in the latter. Consistent with the LC–MS results, no cryo-EM density corresponding to the mcm⁵ group at the U34 base was observed in complete vRNAP (Fig. 7c, red arrow). Indeed, the presence of the mcm⁵s²U34 modification would likely result in unfavorable interactions with the adjacent N-lobe of NPH-I (Fig. 7d, red arrow). We corroborated these results by showing that γ-toxin endonuclease from *Kluyveromyces lactis*[18] failed to cleave complete vRNAP-associated tRNA[Gln/Arg] (Fig. 6e). Therefore, a selective mechanism ensures that only tRNA[Gln/Arg] lacking the mcm⁵s² modification at base U34 can drive the assembly of complete vRNAP.

## Complete vRNAP facilitates en bloc delivery of an entire transcription system

The tRNA[Gln/Arg]-mediated assembly of complete vRNAP ensures the integration of all early transcription and processing factors into one stable unit. We hypothesized that this mechanism evolved to equip virions with precisely those components needed to start transcription upon infection. To test this prediction, purified vaccinia virions were analyzed for their RNA and protein content using RNA sequencing (RNA-seq) and MS, respectively (Fig. 8a,b). The RNA content within the virion cores was of low complexity, with over 60% being tRNA[Gln] and tRNA[Arg] (that is, the tRNA species that facilitate the assembly of complete vRNAP) (Fig. 8a and Supplementary Table 2). Consistent

with this, the proteomics analysis showed that all subunits of complete vRNAP were present in the virion with comparable stoichiometry (Fig. 8b, red dots), whereas intermediate and late transcription factors were either absent or greatly underrepresented (Fig. 8b, green dots, and Supplementary Table 1). Notably, the ratio of tRNA[Gln/Arg] to the vRNAP core subunit Rpo132 was strikingly similar between isolated complete vRNAP and virion core (Extended Data Fig. 9). These findings strongly suggest that complete vRNAP is transferred en bloc into newly formed virions, ensuring that the viral progeny is fully equipped with the machinery needed for early transcription. This notion is further corroborated by the observation that repression of Rap94, which we

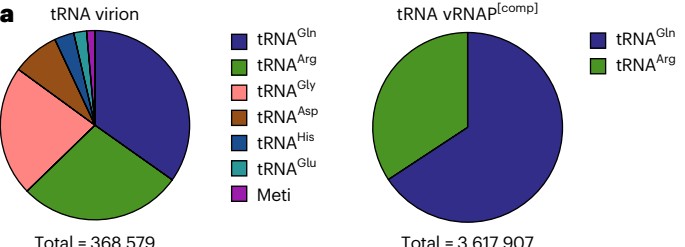

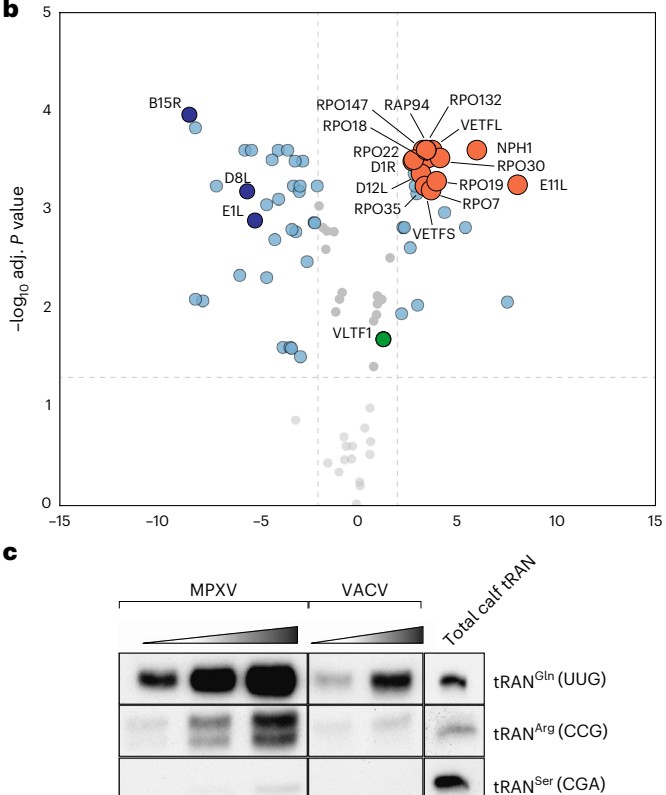

**Fig. 8 | Complete vRNAP is packaged en bloc into vaccinia virions. a**, Quantitative RNA-seq of RNA extracted from vaccinia virions (left) and purified complete vRNAP (right). The total number of reads is indicated below. **b**, Proteomics of purified complete vRNAP compared to total lysate from isolated vaccinia virions. Subunits of complete vRNAP are shown in red; typical abundant proteins of the virion are shown in dark blue. The only detectable intermediate or late transcription factor is shown as a green dot. **c**, Specific incorporation of tRNA[Gln/Arg] into MPXV virions. RNA was extracted from MPXV and vaccinia virions and analyzed by northern blot analysis using probes against tRNA[Gln](UUG), tRNA[Arg](CCG) and tRNA[Ser](CGA). Lanes 1–3 correspond to the amount of tRNA extracted from 2.1 × 10⁶, 4 × 10⁶ and 8 × 10⁶ plaque-forming units of MPXV, respectively. Lanes 4 and 5 correspond to tRNA extracted from 5 × 10⁵ and 1 × 10⁶ plaque-forming units of vaccinia virus, respectively. Lane 6 corresponds to 100 ng of total calf tRNA. The experiment was performed in triplicate.

identify as the initial tRNA[Gln/Arg]-binding platform, results in production of virus particles devoid of vRNAP[19].

Mpox virus (MPXV) is the causative agent of the zoonotic viral disease Mpox[20]. Interestingly, its vRNAP subunits are almost identical to their vaccinia counterparts and would allow a similar tRNA-binding modus to that described for vaccinia complete vRNAP. In accordance with this, we observed a massive enrichment of tRNA[Gln/Arg] in purified MPXV virions (Fig. 8c). This suggests that the formation of a tRNA[Gln/Arg]-induced early transcription unit and its subsequent packaging into virions represent a conserved mechanism among various poxviruses.

## Discussion

In this study, we dissected the entire assembly pathway of complete vRNAP through a combination of biochemical and structural (cryo-EM) approaches. Our findings identify tRNA[Gln/Arg] as assembly chaperones for a macromolecular transcription and processing complex and, thus, add a novel aspect to the functional repertoire of tRNAs beyond their well-established role in translation.

Assembly chaperones are a heterogeneous group of factors that join components with otherwise little or no affinity to each other and typically undergo conformational changes during their action[21]. They have, however, no further role with respect to the function of the assembled unit and often dissociate from the assembled particle. Traditionally, assembly chaperones have been attributed solely to proteins; examples include RbcX and Raf1 in the assembly of the multimeric Rubisco enzyme[22,23] and pICln in the biogenesis of spliceosomal Sm class U snRNPs[24,25]. To the best of our knowledge, tRNA[Gln/Arg] represents the first nonprotein of this group.

We showed how an induced fit mechanism within the ASL facilitates the faithful assembly of a 16-component system. This generates an autonomous transcription unit, which is essential for the replication of poxviruses. Of note, the individual adjunct factors and vRNAP have little if any affinity to each other but essentially require the tRNA for joining. The emerging complete vRNAP complex is robust both in vivo and in vitro but will spontaneously convert to the PIC and initiate transcription when a cognate promoter is present. Remarkably, tRNA[Gln/Arg] establishes a major checkpoint in this transition, impressing in the structure of complete vRNAP as a mechanical (Fig. 4b) and electrostatic (Fig. 4c) barrier for the transcription-factor-bound promoter DNA. Thus, tRNA[Gln/Arg] fulfills all key criteria of an assembly chaperone by facilitating the formation of an otherwise instable active transcription complex without directly participating in its catalytical activity. This role is, hence, mechanistically different from the recently discovered scaffolding function of tRNA[Val] in the mitoribosome[26] and the human cytomegalovirus virion capsid[6].

The role of tRNA[Gln/Arg] extends beyond the prototypical vaccinia virus to the related and pathologically highly relevant MPXV[27]. tRNA[Gln/Arg]-mediated assembly of complete vRNAP may, hence, represent a widespread strategy among poxviruses for incorporating a transiently stalled form of the full early transcription machinery into the virion, thereby providing a convenient mechanism for an immediate initiation of transcription upon infection. It remains to be seen whether RNA chaperones have also evolved in the context of other systems.

Our data show that the modification status of tRNA[Gln/Arg] affects its role in assembly, with $m^1A$ at position 57 and Cm at position 32 being stimulatory. Despite a global upregulation of the $mcm^5s^2U$ modification in total tRNAs upon infection, tRNA[Gln](UUG) carrying this modification in the anticodon would be incompatible with its function in assembly (Fig. 7d). An intriguing question is how tRNA[Gln/Arg] displaying a specific modification pattern (that is, absence of $mcm^5s^2U34$ and presence of $m^1A$ and Cm) is diverted from the host translation machinery into the vRNAP assembly pathway. In uninfected cells, tRNAs are predominantly associated with cellular factors such as their cognate aminoacyl-tRNA synthetases and elongation factors (EFs). In particular, the extremely abundant translation factor EF1α would be expected to be an efficient binding competitor of vRNAP for charged tRNA[Gln/Arg]. However, tRNAGln/Arg associated with vRNAP was shown to be uncharged and is, thus, presumably disengaged from active translation and interactions with the aforementioned cellular factors[8]. This raises the question of how tRNA[Gln/Arg] can be 'stolen' from the host for its activity in complete vRNAP assembly. As fully modified tRNAs are efficiently aminoacylated and likely sequestered by EF1α tRNA[Gln] and tRNA[Arg] lacking $mcm^5s^2U$ may be less aminoacylated, as these hypomodified variants might avoid competition with EF1α and, thus, be preferentially used for vRNAP assembly. This might explain the observation that complete vRNAP contains exclusively uncharged tRNA[Gln/Arg]. An alternative but not mutually exclusive scenario is that a local depletion of glutamine (and arginine) at viral replication sites may drive a shift from charged to uncharged tRNAs. Notably, such a situation may occur in the late phase of infection when complete vRNAP is formed for packaging and may provide a mechanism to link the metabolic status of the infected cell to the viral replication processes. In this regard, it is also noteworthy that glutamine is a crucial metabolite in viral energy metabolism[28,29].

## Online content

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

## Methods

### Purification of vRNAP complexes

Complete and core vRNAP complexes were purified from Hela S3 cells infected with GLV-1h439 virus as described previously[30]. To obtain highly pure complete and core vRNAP samples, FLAG-purified total vRNAP fractions obtained from GLV-1h439-infected HeLa cells were concentrated and separated by a 10–30% sucrose gradient centrifugation step at 164,199g, 16 h and 4 °C. Core vRNAP sedimented under these conditions in fractions 10–13, while complete vRNAP sedimented in fractions 14–16 (Fig. 1b and Extended Data Fig. 1c). The respective fractions were analyzed by denaturing SDS–PAGE, concentrated and used for biochemical and structural studies. For the purification of vaccinia virus minimal RNAP, 370 pmol of intermediate RNAP was mixed with 2.56 nmol of recombinantly expressed VITF-3. Adenosine triphosphate, guanosine triphosphate, cytidine triphosphate and DTT were added to a final concentration of 10 mM each and the sample was incubated for 15 min at 25 °C. Then, 4 nmol of 70-bp G8R promoter DNA scaffold spanning positions −35 to +35 containing a G-less cassette starting at +8 was added and the sample was adjusted to 50 mM HEPES pH 7.5, 150 mM NaCl and 1.5 mM MgCl₂. After 30 min of incubation at 30 °C, the sample was applied on a 5–45% sucrose gradient and ultracentrifuged in a SW60Ti rotor at 1 64,199g for 16 h at 4 °C. The gradient was isolated in 200-μl fractions; fractions 10–14 were pooled and concentrated in a Vivaspin 500 (100-kDa molecular weight cutoff (MWCO); Satorius, VS0141) to 2.4 μg μl⁻¹.

### In vitro reconstitution of vRNAP assembly intermediates

Isolated tRNA$^{Gln/Arg}$ was 3′-end-labeled with [$^{32}$P]pCp according to the manufacturer's protocol (Thermo Fischer Scientific, EL0021). Labeled RNA was separated by denaturing RNA gel electrophoresis (10%, 19:1, 8 M Urea, TBE), eluted from the gel in TBE and precipitated with ethanol. The pellet was washed with 70% ethanol and resuspended in resuspension buffer (10 mM Tris-HCl pH 8.0 and 1 mM EDTA). For the vRNAP reconstitution assays shown in Fig. 1b, 40 pmol of minimal, core or complete vRNAP was incubated with 80 pmol of [$^{32}$P]tRNA at 30 °C for 30 min. After incubation mixtures were separated by density gradient centrifugation (10–30% sucrose gradient, 164,199g for 16 h, at 4 °C). Gradient fractions were collected manually and analyzed by SDS–PAGE and RNA gels. RNA was visualized by autoradiography and proteins were visualized by silver staining. For reconstitution of higher-order assembled intermediates (Fig. 1c), 1 pmol of [$^{32}$P]pCp-labeled tRNA$^{Gln/Arg}$ isolated from complete vRNAP was incubated with 2.4 pmol of core vRNAP in the presence of recombinant factors (D1/D12, E11 and NPH-I). After incubation for 30 min at 25 °C, samples were analyzed by native electrophoresis (4% acrylamide, 0.13% bisacrylamide, 25 mM Tris-HCl pH 7.4, 25 mM boric acid and 0.5 mM EDTA). Native electrophoresis was performed at 4 °C and complexes were visualized by autoradiography. For large-scale reconstitution of assembly intermediate I3, 20 mg of core vRNAP (40 pmol) was incubated with 10 mg of tRNA$^{Gln/Arg}$ (400 pmol), recombinant E11 (900 pmol) and NPH-I (900 pmol) for 30 min at 25 °C. Reconstituted I3 was purified by anti-FLAG affinity chromatography. For cryo-EM analysis, FLAG peptide was removed by sample concentration using Vivaspin (Satorius, VS0101) with a 10-kDa MWCO.

To demonstrate that individual protein factors have no affinity to core vRNAP, recombinant NPH-I and D1/D12 or E11 were incubated with core vRNAP (molar ratio of 10:10:100:1) for 30 min at 25 °C. The mixture was subsequently separated by a 10–30% density gradient by centrifugation step (164,199g for 16 h and 4 °C). Sucrose fractions were collected manually and separated by 12% Bis–Tris gel electrophoresis[30]; proteins were visualized by silver staining. The interactions among recombinant factors (E11, NPH-I and D1/D12) were investigated by coimmunoprecipitation. Then, 10 μl of Dynabeads protein G was incubated with an affinity-purified monospecific anti-NPH-I antibody for 1 h at 25 °C. For control experiments, protein G beads were incubated with PBS buffer only. The beads were subsequently washed three times with PBS pH 7.5 and mixed with recombinant factors for 30 min at 25 °C in buffer containing 150 mM NaCl, 50 mM HEPES pH 7.5, 1.5 mM MgCl₂ and 1 mM DTT. Beads were washed three times with buffer containing 250 mM NaCl, 50 mM HEPES pH 7.5, 1.5 mM MgCl₂ and 1 mM DTT and bound proteins were analyzed by SDS–PAGE.

### tRNA isolation

tRNA$^{Gln/Arg}$ was obtained from affinity-purified complete vRNAP by TRIzol extraction (Invitrogen, 15596026). For isolation of total tRNA, Hela S3 cells were either mock-infected or infected with GLV-1h439. Cells were harvested and TRIzol was directly added to cell pellets for lysis. The aqueous phase was precipitated with ethanol and resuspended in TBE; total RNA was separated on denaturing RNA gels (8%, 19:1 acrylamide:bisacrylamide, 8 M urea and TBE). tRNAs were visualized by ethidium bromide and eluted from the gel in TBE buffer. Eluted tRNAs were precipitated with ethanol and resuspended in water or resuspension buffer (10 mM Tris-HCl pH 8.0 and 1 mM EDTA). Total calf tRNA was obtained from (Roche, 83298320-63). Wild-type tRNA$^{Gln}$ and a variant mutated at position U33G were synthetized by T7 transcription with a high-yield T7 transcription kit (Thermo Fischer Scientific, K0441) according to the manufacturer's protocol.

### Recombinant *Escherichia coli* expression of D1/D12, NPH-I and *K. lactis* γ-toxin

For the recombinant expression of vaccinia virus capping enzyme D1/D12 and early transcription termination factor NPH-I, the respective open reading frames were amplified from vaccinia virus genomic DNA and cloned into pET-Duet1 or pETM-11 vectors to generate pET-Duet1–D1-D12 and pETM-11-NPH-I, respectively. The resulting plasmids were verified by sequencing and used for protein expression experiments. A plasmid allowing for the expression of GST−γ-toxin was kindly provided by R. Schaffrath and D. Scherf[31]. Expression and purification of E11 was performed as described previously[8]. For purification of the His−D1/D12 heterodimer and His−NPH-I, pETDuet-D1R-D12L or pETM-11-NPH-I is transformed into *E. coli* Iq cells (New England Biolabs, C3037l). *E. coli* was grown in SB medium containing ampicillin and chloramphenicol for PETDuet-D1R-D12 or kanamycin and chloramphenicol in a case of pETM-11-NPH-I. At an optical density of 0.8, protein expression was induced by adding 0.5 mM IPTG for 16 h at 16 °C. Cells were pelleted by centrifugation and lysed by sonication in buffer containing 150 mM NaCl, 50 mM HEPES pH 8.0, 15 mM imidazole, 2 mM β-mercaptoethanol and protease inhibitors. The cleared lysate was mixed with 1 ml of Ni-NTA beads (His60 Ni Superflow Resin; Takara, 635660) and incubated at 4 °C for 3 h. The beads were washed three times with washing buffer (150 mM NaCl, 50 mM HEPES pH 8.0, 25 mM imidazole and 2 mM β-mercaptoethanol) and proteins were eluted in washing buffer containing 250 mM imidazole. The eluates were applied to a Superdex 200 (HiLoad 26/600 PG) column and analyzed by SDS–PAGE. Pure fractions eluting at the expected volume were collected, concentrated in Vivaspin 6 (10-kDa MWCO; Satorius, VS0101) to 3 μg μl⁻¹ for NPH-I and 1.5 μg μl⁻¹ for D1R/D12L and used for biochemical reconstitution assays (Extended Data Fig. 1a–c and Figs. 1c and 2).

GST−γ-toxin was expressed in the *E. coli* BL21(DE3) pRARE strain. Expression was induced with 0.5 mM IPTG (final concentration) for 16 h at 18 °C. Cells were collected at 4,000g at 4 °C and resuspended in lysis buffer (50 mM Tris-HCl pH 7.5, 300 mM NaCl, 2 mM DTT and 1 mg ml⁻¹ lysozyme). After sonication, the lysate was cleared and the supernatant was incubated with 10 ml of glutathione Sepharose beads (Cytiva, 17513202). The column was washed three times with lysis buffer omitting lysozyme and the protein was eluted in the same buffer containing 20 mM glutathione. The elution was applied onto a Superdex 200 column (HiLoad 26/600 PG). GST−γ-toxin eluted as a single peak and was concentrated in Vivaspin 6 (10-kDa MWCO; Sartorious, VS0101) to 2.5 μg μl⁻¹ before it was used for RNA cleavage assays.

## tRNA cleavage assay with γ-toxin and northern blot analysis

To investigate the mcm$^5$s$^2$U34 modification of tRNA associated with complete vRNAP, a cleavage assay using γ-toxin was performed as described previously[8]. vRNAP-associated tRNA purified by TRIzol extraction and total calf tRNA were 3′-end-labeled with [$^{32}$P]pCp and purified by denaturing gel electrophoresis. Then, 100 ng of isolated tRNAs were incubated with buffer (20 mM HEPES pH 7.5, 150 mM NaCl, 2 mM EDTA and 2 mM DTT) or increasing amounts of γ-toxin in a total volume of 15 μl at 30 °C for 30 min. tRNA was subsequently TRIzol-extracted, separated by denaturing RNA gel electrophoresis and visualized by autoradiography.

For analysis of tRNAs from Hela S3 cells infected with vaccinia virus, total tRNAs were purified as described above. Then, 100 ng of total tRNAs collected from infected cells were digested with an increasing amount of γ-toxin, followed by TRIzol extraction. Full-length tRNAs and digested fragments were separated on denaturing RNA gels and transferred to a nylon membrane (transfer at 72 mA for 1 h). The membrane was hybridized with a $^{32}$P-labeled DNA probe (5′-AGGTCCCACCGAGATTTGAACTCG-3′) complementary to the 3′-end of tRNA$^{Gln}$(UUG) at 42 °C for 12 h. The membrane was washed three times with NB washing buffer (2× SSC and 0.01% SDS) and bands were visualized by autoradiography.

For identification of tRNA$^{Gln/Arg}$ in MPox and vaccinia virus, RNA was purified by TRIzol extractions from virions, precipitated and resuspended in RNAase-free water. For northern blot analysis, RNA was separated in a 12% denaturing RNA gel and transferred to a nylon membrane (GE Healthcare, RPN203B) at 72 mA for 1 h. After transfer, the membrane was hybridized at 42 °C with 5′-$^{32}$P-labeled DNA oligonucleotides (5′-ACTCGGATCGCTGGATTCAAAGTCCAGAGTGCTA ACCA-3′, 5′-ACCCTCAATCTTCTGATCCGGAATCAGACGCCTTATCCA-3′ or 5′-CGGGGAGACCCCATTGGATTTCGAGTCCAACGCCTTAACC ACT-3′) complementary to the anticodon regions of tRNA$^{Gln}$(UUG), tRNA$^{Arg}$(CCG) and tRNA$^{Ser}$ (CGA), respectively. The membrane was washed three times with a washing buffer and visualized by autoradiography as described above.

## Cryo-EM structure determination and model building of vaccinia complexes

The sucrose-gradient-purified vRNAP samples prepared as described above were diluted 1:50 and concentrated in a Vivaspin concentrator to a concentration of roughly 1 mg ml$^{-1}$ to remove the sucrose, centrifuged for 2 h at 21,000g and diluted 1:1 in a buffer containing 20 mM HEPES pH 7.5, 200 mM (NH$_4$)$_2$SO$_4$, 1 mM MgCl$_2$ and 5 mM β-mercaptoethanol. Next, R 1.2/1.3 Quantifoil holey carbon grids (Jena Bioscience) were glow discharged for 90 s in a plasma cleaner (PDC-002, Harrick Plasma) at medium power and 3.5 ml of C2 sample was applied inside a Vitrobot Mark IV (Thermo Fisher Scientific) at 4 °C and 100% relative humidity. The grids were blotted for 3 s with a blot force of 5 and plunged into liquid ethane. The cryo-EM datasets were collected with Titan Krios G3 equipped with a Falcon III or Falcon IV camera (Thermo Fischer Scientific). Data were acquired with EPU at 300 keV.

Datasets were processed with cryoSPARC[32]. Several cycles of two-dimensional classification and manual selection of classes based on the appearance of their class averages were applied for initial cleanup. Ab initio maps were created from a subset of 50,000 particles and the full set was subjected to a consensus three-dimensional refinement. The datasets were further classified as detailed in Extended Data Figs. 2 and 4–6. The classified particle sets were finally subjected to a nonuniform refinement step[33] and the density was docked with initial models derived manually from the previously published complete vRNAP model (PDB 6RFL). In a first round, each model was manually refined including removal of some stretches of Rap94 that were not represented in the cryo-EM density. The models were then subjected to alternating rounds of automatic refinement with phenix.real_space_refine[34] including an atomic displacement parameter refinement step and manual building with Coot[35]. During automated refinement, secondary-structure and mild Ramachandran restraints were imposed. A total of three cycles of manual inspection and automated refinement were performed for each model.

## MS

In-solution digestion: For the analysis of vRNAP and virion samples by MS, proteins were precipitated with 80% acetone at −20 °C for 2 h. Proteins were resolubilized in 8 M urea and 50 mM ammonium bicarbonate to reach a concentration of 0.5 μg μl$^{-1}$ followed by reduction with 5 mM Tris(2-carboxyethyl)phosphine dissolved in 10 mM ammonium bicarbonate (30 min at 56 °C) and alkylation of free thiol groups with 100 mM 2-chloracetamide and 10 mM ammonium bicarbonate (30 min at 37 °C). Urea concentration was adjusted to 2 M by adding 50 mM ammonium bicarbonate, trypsin was added at a protease-to-protein ratio of 1:50 and samples were incubated for 16 h at 37 °C and 200 rpm. Protein digestion was stopped by acidification with trifluoroacetic acid (TFA) added at a final concentration of 1% (v/v). Peptides were dried in vacuo and stored at −80 °C until used for LC–MS analysis.

LC–MS analysis: Dried peptides were reconstituted in 0.1% (v/v) TFA and analyzed by reverse-phase LC–MS using an UltiMate 3000 RSLCnano system (Thermo Fisher Scientific) coupled online to a Q Exactive Plus instrument (Thermo Fisher Scientific). The LC system was equipped with C18 precolumns (μPAC trapping column, PharmaFluidics) and a C18 endcapped analytical column (50-cm μPAC column, PharmaFluidics). Peptide separation and elution were performed at 40 °C using a binary solvent system composed of 0.1% (v/v) formic acid (FA) (solvent A) and 86% (v/v) acetonitrile in 0.1% (v/v) FA (solvent B). Peptide mixtures from RPO132 pulldown experiments were analyzed by LC–MS using a 3-h method. Peptides were loaded at 1% solvent B for 3 min at a flow rate of 10 μl min$^{-1}$ and eluted by applying the following gradient: 5–22% B in 96 min, 22–42% B in 54 min, 4 min at 80% B and re-equilibration of the column for 19 min at 100% A. The flow rate for peptide elution was set to 0.3 μl min$^{-1}$. The MS instrument, equipped with a nanoelectrospray ion source and a stainless-steel emitter (Thermo Fischer Scientific), was externally calibrated using standard compounds. Parameters for MS measurements in data-dependent acquisition mode were as follows: MS full scan window of m/z 375–1,700, resolution of 70,000 (at m/z 200), automatic gain control (AGC) of $3 × 10^6$ and maximum injection time (IT) of 60 ms. Multiply charged peptide ions were fragmented by higher-energy collisional dissociation applying a normalized collision energy of 28% and a dynamic exclusion time of 45 s. A TOP12 method was applied to analyze peptides with an MS2 resolution of 35,000, an AGC of $7 × 10^2$ and a maximum IT of 120 ms.

MS data analysis: MS raw data were processed using MaxQuant/Andromeda (version 2.0.2.0)[36] and the vaccinia virus (strain Western Reserve) reference proteome from UniProt (UP000000344; downloaded January 2022, containing 218 protein entries). Proteins were identified using MaxQuant default settings. Oxidation of methionine and N-terminal acetylation were considered as variable modifications, while carbamidomethylation of cysteine residues was set as the fixed modification. The option 'match between runs' was enabled.

Bioinformatic analysis: For the identification of proteins enriched in vaccinia virus virions, data were first normalized as follows. For each replicate, the summed MS intensity of all protein groups was shifted to the mean of the summed MS intensities determined across all three replicates. Subsequently, intensities were transformed by variance stabilizing data transformation[37]. Protein abundance ratios were determined and the linear models for microarray data (limma) approach, as implemented in the R package limma (version 3.60.4)[38,39], was applied to calculate P values for the enrichment of proteins.

## Analysis of the tRNA$^{Gln/Arg}$ modification by LC–MS

Extracted tRNA$^{Gln/Arg}$ from the vRNAP complex was purified with an RNA clean and concentrator-5 (Zymo Research, lot 213186). Then, 400 pmol

of tRNA was digested by 6.0 U of bacterial alkaline phosphatase and 1.0 U of snake venom phosphodiesterase in reaction buffer (40 mM Tris-HCl and 20 mM MgCl$_2$, pH 7.5). After extracting the digested nucleosides mixture with 100 µl of chloroform, the aqueous layer was concentrated by lyophilization and the residue was dissolved in 70 µl of 10 mM ammonium acetate. The analysis was run with a gradient of 0–5% (0–15 min) and 5–72.5% (15–45 min) of solvent B, using an Synergi Fusion RP column (Phenomenex; 4 µm, 250 × 2 mm). Solvent A was 10 mM ammonium acetate (pH 5.3), solvent B was acetonitrile and the flow rate was 0.2 ml min$^{-1}$ at 25 °C with ultraviolet detection at 260 nm and online MS in a microTOF-Q III system in positive ion mode. The same experiment was also performed for in vitro transcription using tRNA$^{Gln}$ as control and for total tRNA extracted from infected and noninfected cells.

### RNA preparation for sequencing

Libraries were generated from the isolated RNA fraction following the Ion Torrent ion total RNA-seq kit v2 (Thermo Fisher Scientific, 4475936) protocol with some modifications. Briefly, 30 ng of the RNA was incubated with 5 U of RNase T1 (Thermo Fisher Scientific, EN0541) at 20 °C for 35 min. The samples were then treated with 5 U of Antarctic phosphatase (New England Biolabs, M0289) for 30 min at 37 °C. After heat inactivation at 65 °C, the RNA was phosphorylated with 20 U of T4 polynucleotide kinase (New England Biolabs, M0201) for 60 min at 37 °C. Adaptor ligation was performed for 16 h at 16 °C. Reverse transcription (RT) was performed using SuperScript III with incubations at 42 °C, 50 °C and 55 °C for 45, 15 and 10 min, respectively. The RT reactions were purified and the complementary DNA was amplified by Platinum PCR SuperMix high fidelity. The resulting libraries were sequenced using an Ion Proton (Ion Torrent TM) with Hi-Q.

### Reporting summary

Further information on research design is available in the Nature Portfolio Reporting Summary linked to this article.

## Data availability

Data are available in the main text or the Supplementary Information. PDB validation reports were deposited to figshare or are freely available from the PDB. Cryo-EM maps of complete vRNAP, minimal vRNAP and all assembly intermediates newly described in this work were deposited to the EM Data Bank. Atomic coordinates of the corresponding models are available from the PDB. EMD accession codes and PDB accession codes are as follows: I5, EMD-16476 and PDB 8C8H; complete vRNAP, EMD-19442 and PDB 8RQK; I3, EMD-50639 and PDB 9FPY; I4, EMD-50644 and PDB 9FQ6; minimal vRNAP, EMD-50033 and PDB 9EX9. All other data not publicly available at the time of the review of the manuscript were deposited to figshare for review purposes. The proteomics raw data can be accessed through the PRIDE website under accession PXD057359. Source data are provided with this paper.

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

## Acknowledgements

We thank S. Leidel, B. Böttcher and C. Kraft for helpful discussions and technical advice and D. Scherf and R. Schaffrath for providing expression plasmid for γ-toxin. This work was supported by German Research Foundation grants to U.F. (Fi-573/23), C.H. (project No 469281184, SFB 1565, P18) and B.W. (FOR 2743), the Volkswagenstiftung to U.F. and C.G. (grant 9B813) and by Austrian Science Fund grants to M.E. (FWF P34132-B, F8004) and A.H. (FWF P32612-B and FWF TAI 411-B). Cryo-EM was carried out in the cryo-EM facility of the Julius Maximilian University of Würzburg funded by the German Research Foundation (projects INST 93/903-1 359471283, INST 93/1042-1 456578072 and INST 93/1143-1 525040890).

## Author contributions

Conceptualization, U.F., C.G. and J.B. Complex reconstitution and purification, J.B. and S.J. Biochemical assays, J.B., M.D. and A.V. MS, B.W. and J.P.Z. tRNA sequence and modification analysis, T.P., A.H., M.E., C.H. and T.O. Cryo-EM sample preparation and data collection, C.G., J.B. and S.J. Data processing, model building, structure analysis and visualization, C.G. Writing, U.F., C.G. and J.B. Funding acquisition, U.F., C.G., M.E., A.H., C.H. and B.W.

## Funding

## Competing interests

The authors declare no competing interests.

## Additional information

**Extended data** is available for this paper at https://doi.org/10.1038/s41594-025-01653-y.

**Correspondence and requests for materials** should be addressed to Clemens Grimm or Utz Fischer.

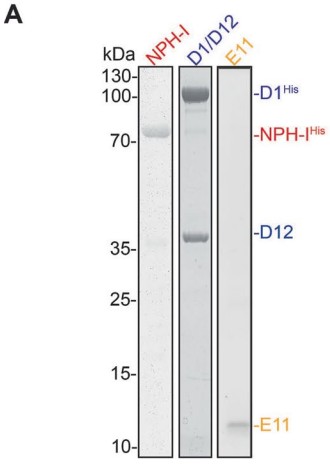

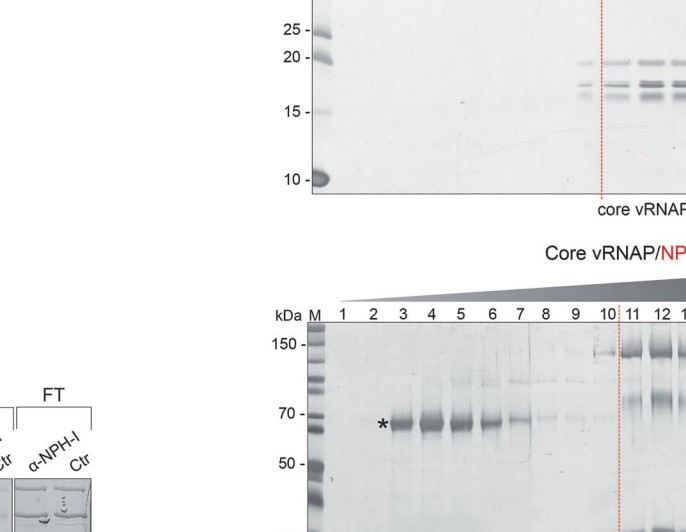

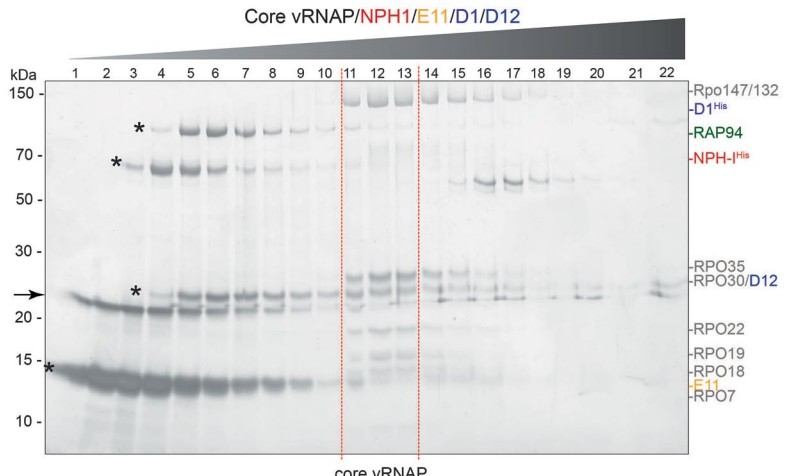

**Extended Data Fig. 1 | Interaction of individual transcription factors with each other and with core vRNAP. (a)** SDS-gel of affinity purified adjunct factors D1/D12, NPH-I and E11 obtained by *E. coli* overexpression and used for reconstitution assays. The arrow indicates an NPH-I degradation product. The experiment was performed in duplicates. **(b)** Minimal binding of E11, D1/D12 and NPH-I to each other. Recombinant NPH-I and E11 (lanes 1-4) or NPH-I, D1/D12 and E11 (lanes 5-8) were incubated with each other and immunoprecipitated by antisera against NPH-I or a non-immune serum. The immunoprecipitates (lanes 1, 2, 5, 6) and supernatants (lanes 3, 4, 7 and 8) were resolved by SDS-PAGE; HC and LC indicate heavy and light chain of the co-eluted antibody, respectively. No co-immunoprecipitation was observed for NPH-I to E11, and only marginal

binding of D1/D12 to NPH-I/E11 was detectable. The arrow indicates an NPH-I degradation product. **(c)** Sucrose density gradient of vRNAP affinity-purified from infected cells (upper panel). Fractions 11-13 contain core vRNAP that was used for reconstitution assays, fractions 15-18 contain complete vRNAP. Gradient centrifugation of core vRNAP (fractions 11-13) pre-incubated with a mixture of either recombinant NPH-I and E11 or D1/D12. None of the recombinant factors bound to a significant amount to the vRNAP (fractions 11-13) but remained at the top of the gradient (fractions 1-6). The arrow indicates an NPH-I degradation product. Asterisks indicate recombinant D1/D12, NPH-I and E11. Note that D12 and Rpo30 migrate at the same position in the gel. The experiment was performed in duplicates.

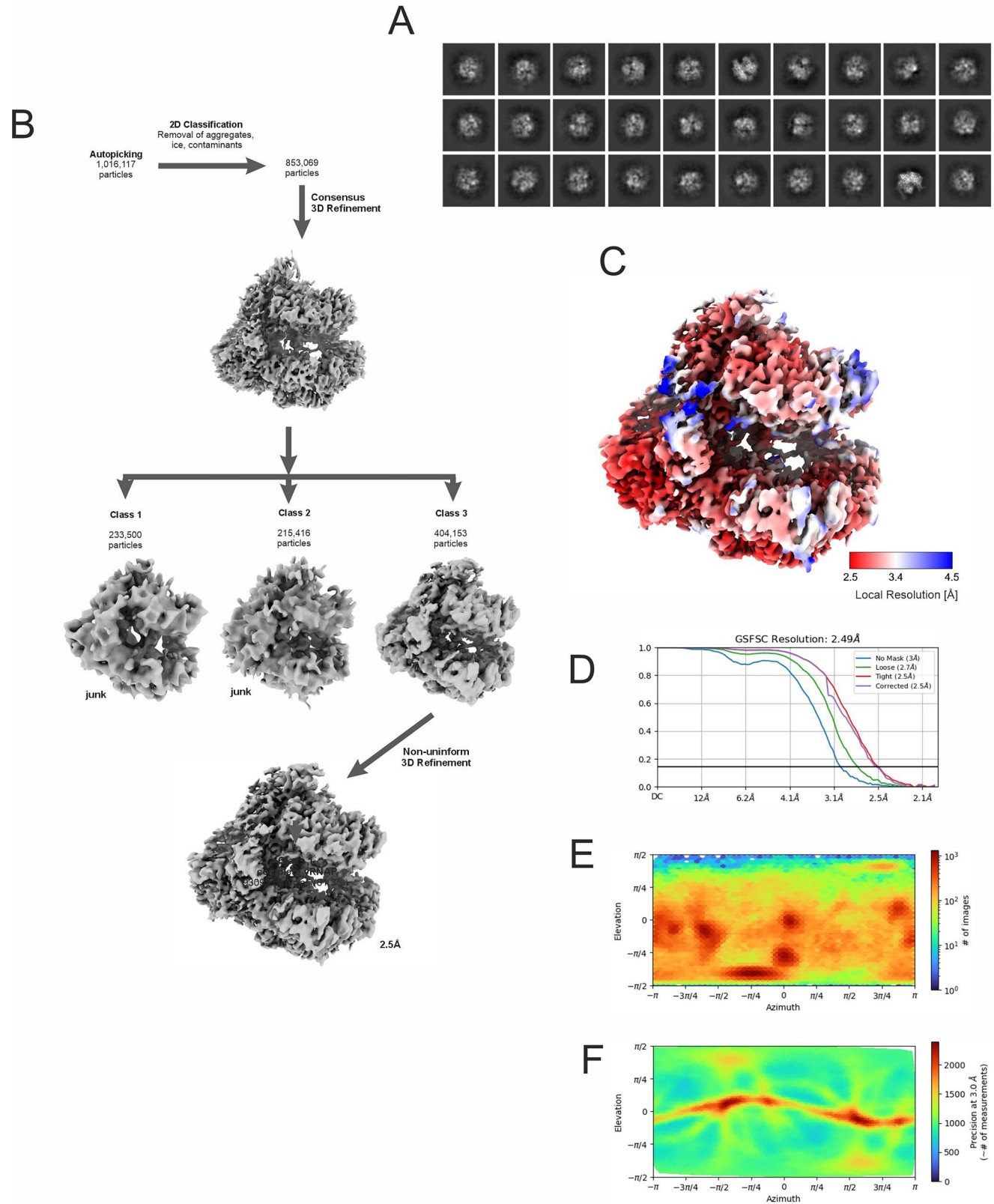

**Extended Data Fig. 2 | Cryo-EM reconstruction of minimal vRNAP obtained from infected cells treated with the replication inhibitor AraC. (a)** Selected 2D classes. **(b)** classification and refinement scheme. **(c)** Local resolution mapped to the reconstruction density isosurface. **(d)** FSC plots for final reconstruction. **(e)** Orientation distribution plot for final reconstruction. **(f)** Posterior precision distribution plot for final reconstruction.

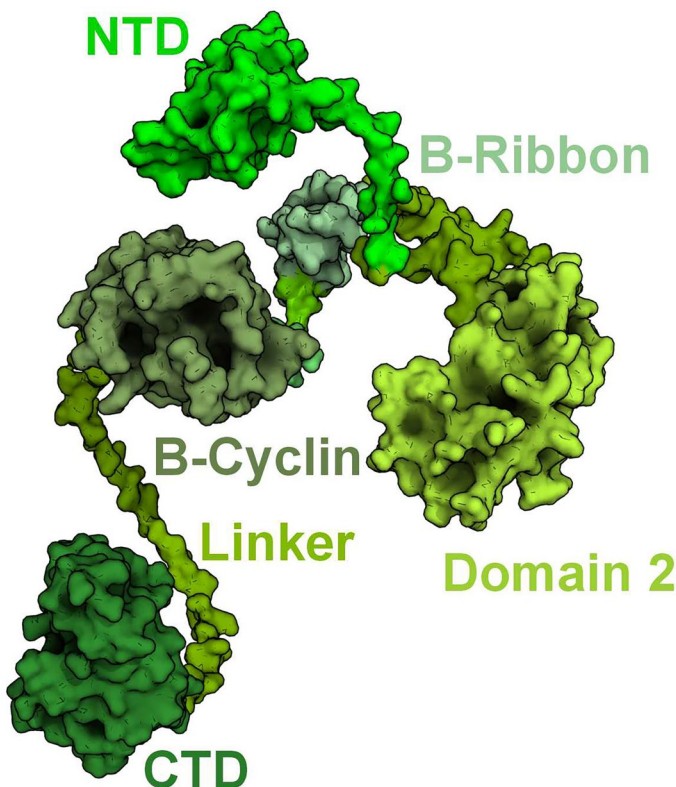

**Extended Data Fig. 3 | Domain arrangement of bound Rap94.** The Rap94 structure as extracted from complete vRNAP is shown in surface representation. The five domains (NTD, Domain 2, B Ribbon, B-cyclin domain and CTD) are coloured in shades of green. The domains are connected by flexible linkers that are fixed on the vRNAP surface throughout the assembly pathway.

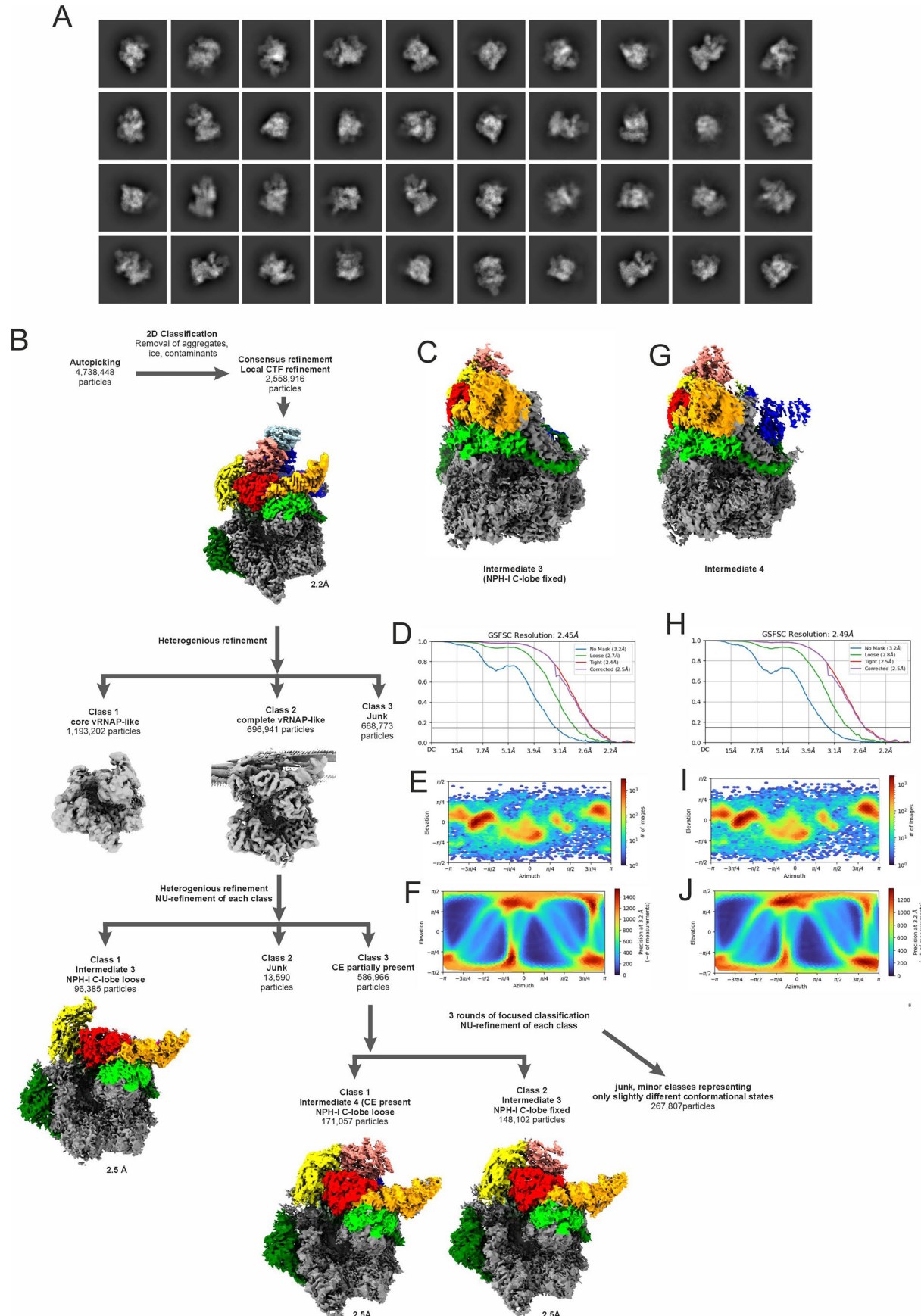

**Extended Data Fig. 4 | See next page for caption.**

**Extended Data Fig. 4 | Cryo-EM reconstruction of vRNAP assembly intermediates I3 and I4. (a)** Selected 2D classes. **(b)** classification and refinement scheme. Domains are colored according to the scheme used in Fig. 2. **(c)** Final reconstruction for intermediate I3. **(d)** FSC plots for final reconstruction of intermediate I3. **(e)** Orientation distribution plot for final reconstruction of intermediate I3. **(f)** Posterior precision distribution plot for final reconstruction of intermediate I3. **(g)** Final reconstruction for intermediate I4. **(h)** FSC plots for final reconstruction of intermediate I4. **(i)** Orientation distribution plot for final reconstruction of intermediate I4. **(j)** Posterior precision distribution plot for final reconstruction of intermediate I4.

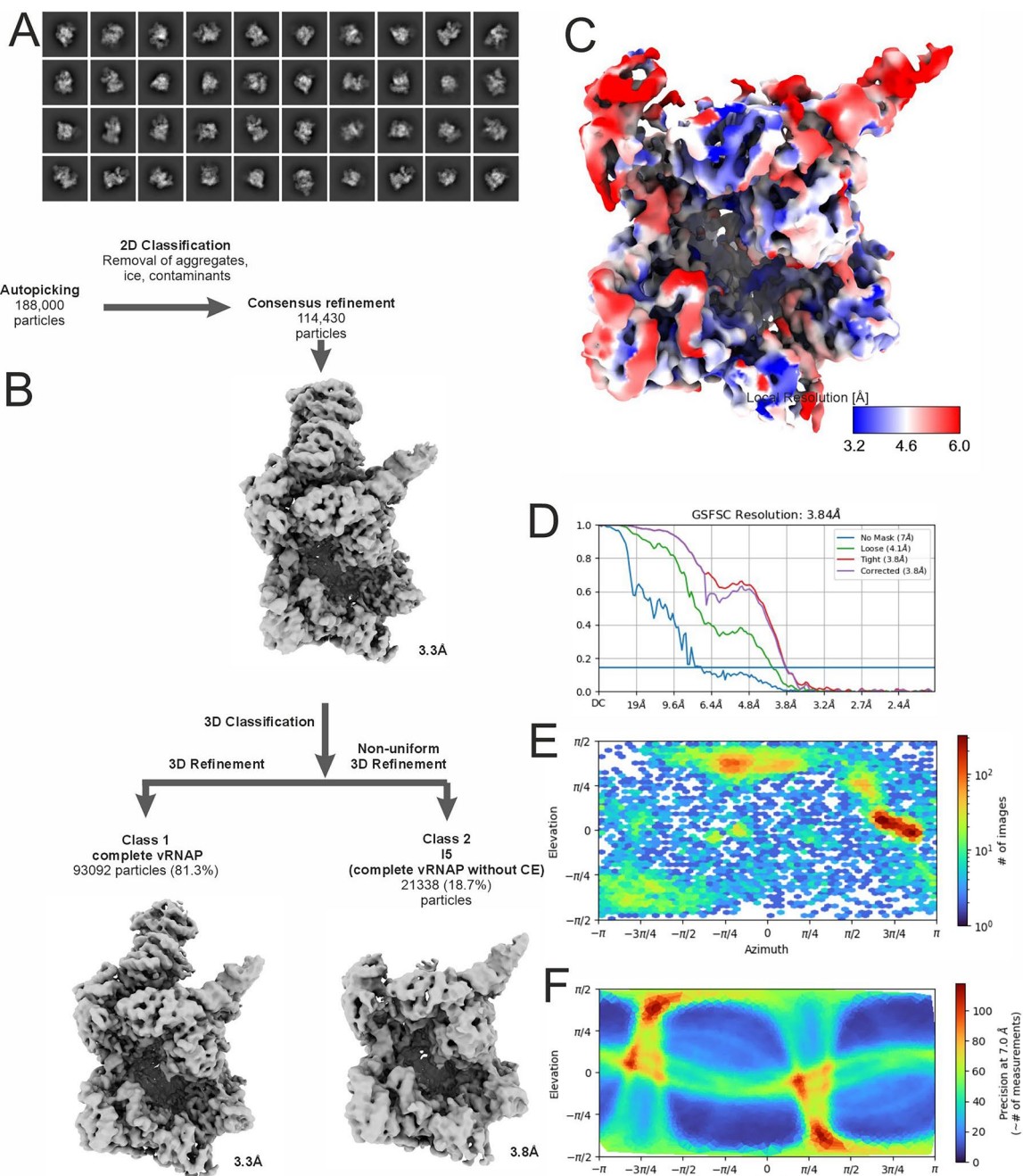

**Extended Data Fig. 5 | Cryo-EM reconstruction of Intermediate 5 (Complete vRNAP lacking the capping enzyme). (a)** Selected 2D classes. **(b)** Classification and refinement scheme. **(c)** Local resolution mapped to the reconstruction density isosurface. **(d)** FSC plots for final reconstruction. **(e)** Orientation distribution plot for final reconstruction. **(f)** Posterior precision distribution plot for final reconstruction.

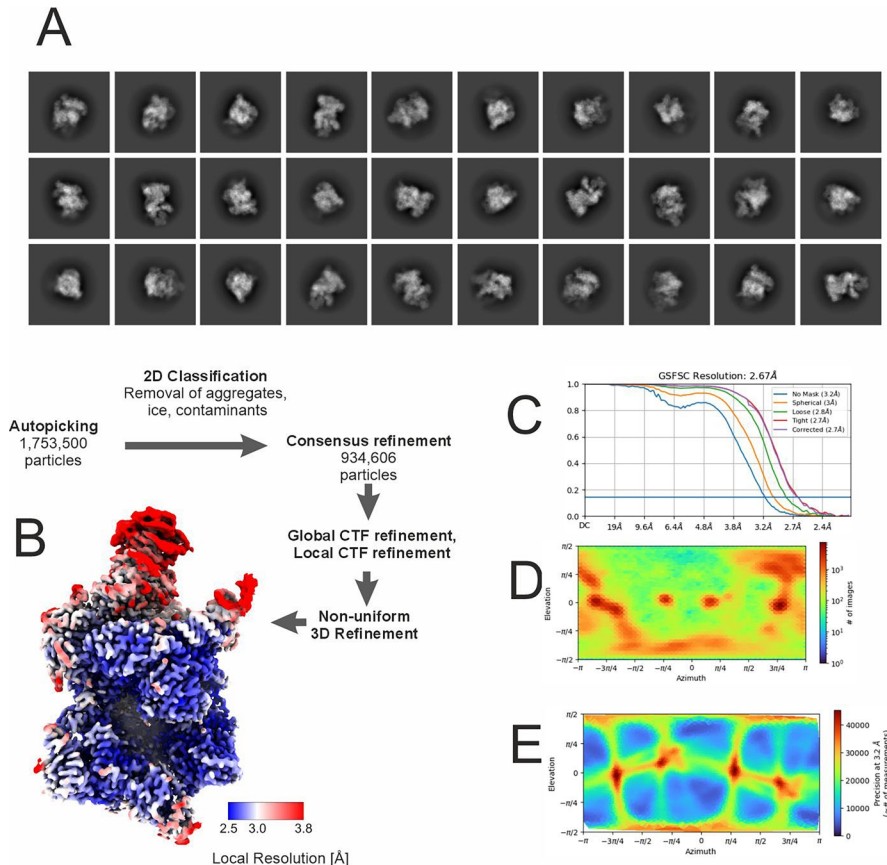

**Extended Data Fig. 6 | Cryo-EM reconstruction of Complete vRNAP. (a)** Selected 2D classes. **(b)** Classification and refinement scheme. The cryo EM isosorface of the final reconstruction is colored by local resolution. **(c)** FSC plots for final reconstruction. **(d)** Orientation distribution plot for final reconstruction. **(e)** Posterior precision distribution plot for final reconstruction.

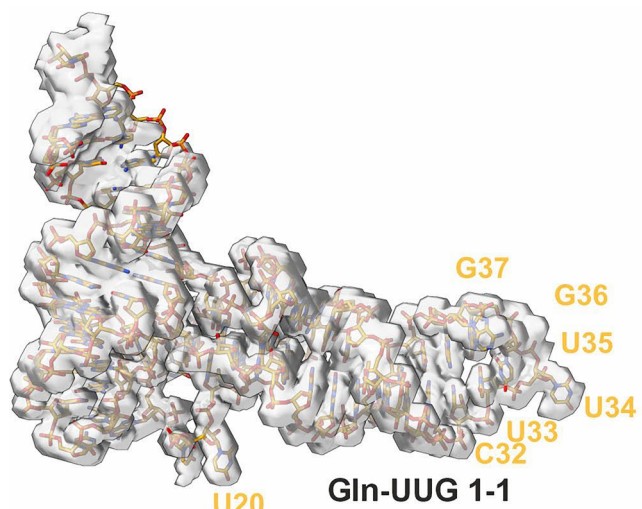

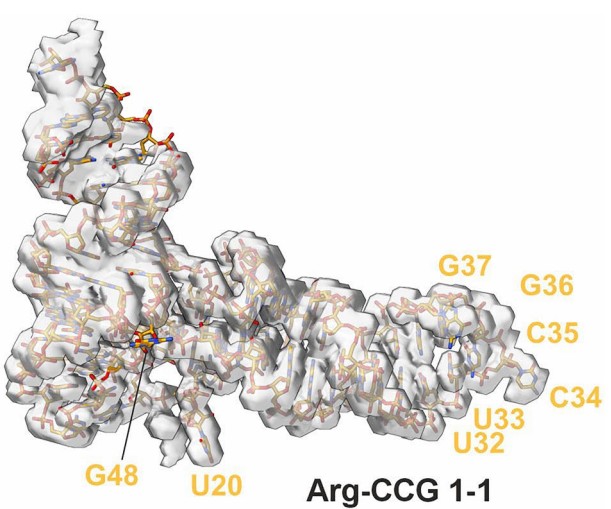

**Extended Data Fig. 7 | tRNA^Gln isoforms with their associated cryo EM density.** Comparison of the tRNA^Gln-UUG model with its associated cryo EM density as extracted from the complete vRNAP structure (left) to a model for the tRNA^Arg-CCG sequence, modelled manually into the complete vRNAP density (right). Both sequences fit the density well, except for the "extra" base G48 in the tRNA^Arg-CCG sequence.

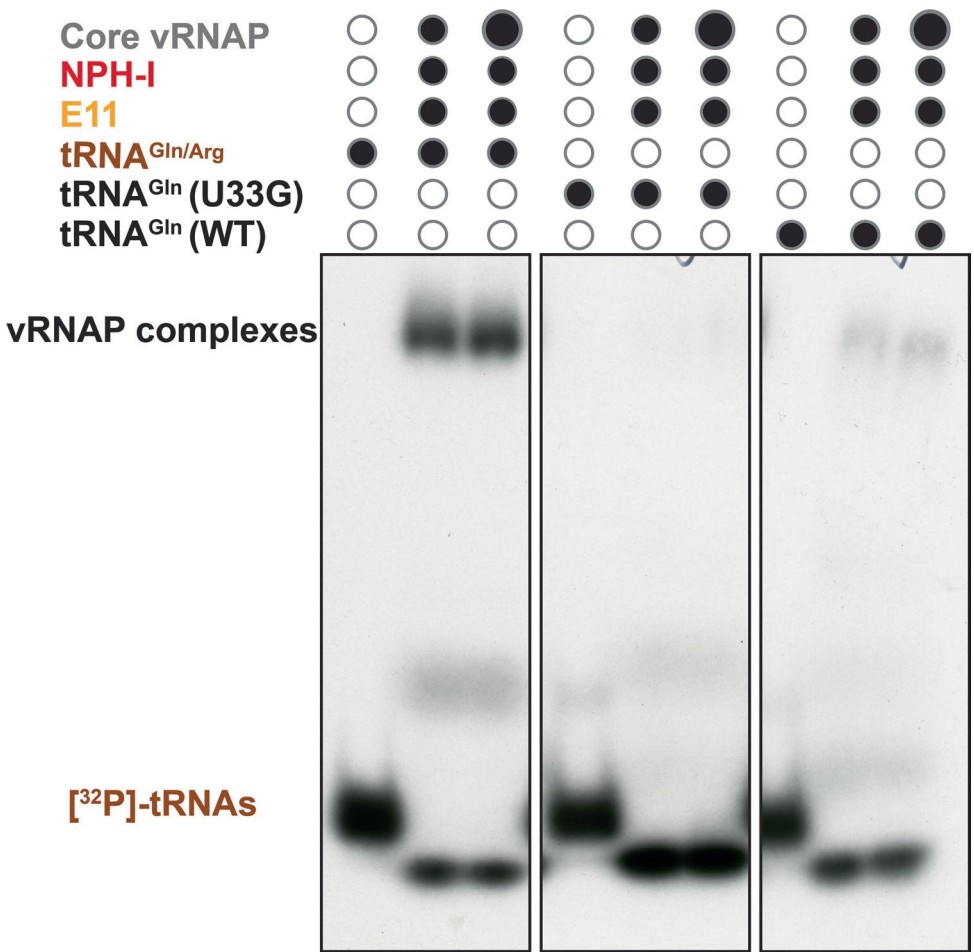

**Extended Data Fig. 8 | Impact of modification and non-canonical base pairing on vRNAP complex formation.** Core vRNAP and the indicated recombinant transcription factors were incubated with [$^{32}$P]-labelled tRNA$^{Gln/Arg}$ (isolated from complete vRNAP), in vitro-transcribed tRNA$^{Gln}$ harbouring a U33G mutation, and in vitro-transcribed wild-type tRNA$^{Gln}$ (WT). Samples were analysed by native gel electrophoresis and complexes were visualized by radioautography. Black circles indicate components that were added, while white circles indicate components that were omitted. The size of the circle reflects the relative amount of each recombinant factor included. The experiment was performed three independent times with similar results.

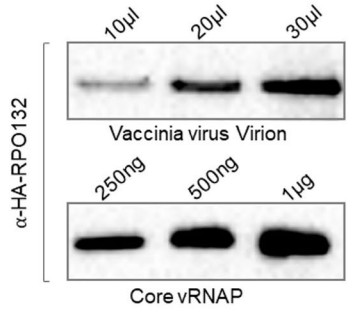

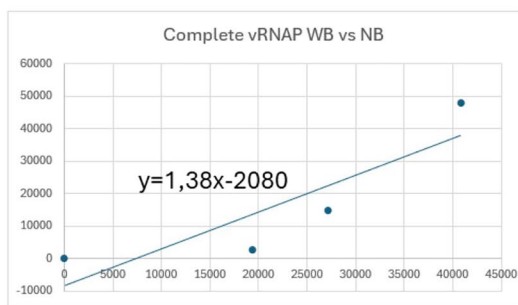

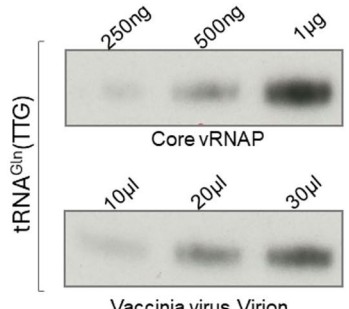

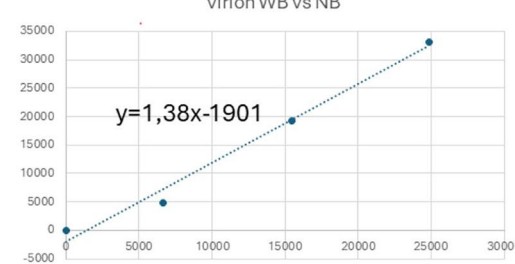

**Extended Data Fig. 9 | Amount of vRNAP and tRNA$^{Gln}$(UUG) contained in purified complete vRNAP and virion extract.** Anti-HA Western blot of isolated complete vRNAP and virions (above left). Northern blot of tRNA$^{Gln}$(UUG) isolated from complete vRNAP and virions (below left). Correlation of the respective Western blot (WB) signals to Northern blot (NB) signals. The experiment was performed in triplicates.

|---|---|
| | Utz Fischer |

# Reporting Summary

## Statistics

For all statistical analyses, confirm that the following items are present in the figure legend, table legend, main text, or Methods section.

| n/a | Confirmed | |
|---|---|---|
| ☐ | ☒ | The exact sample size (*n*) for each experimental group/condition, given as a discrete number and unit of measurement |
| ☐ | ☒ | A statement on whether measurements were taken from distinct samples or whether the same sample was measured repeatedly |
| ☐ | ☒ | The statistical test(s) used AND whether they are one- or two-sided<br>*Only common tests should be described solely by name; describe more complex techniques in the Methods section.* |
| ☒ | ☐ | A description of all covariates tested |
| ☒ | ☐ | A description of any assumptions or corrections, such as tests of normality and adjustment for multiple comparisons |
| ☒ | ☐ | A full description of the statistical parameters including central tendency (e.g. means) or other basic estimates (e.g. regression coefficient) AND variation (e.g. standard deviation) or associated estimates of uncertainty (e.g. confidence intervals) |
| ☐ | ☐ | For null hypothesis testing, the test statistic (e.g. $F$, $t$, $r$) with confidence intervals, effect sizes, degrees of freedom and $P$ value noted<br>*Give P values as exact values whenever suitable.* |
| ☐ | ☐ | For Bayesian analysis, information on the choice of priors and Markov chain Monte Carlo settings |
| ☐ | ☐ | For hierarchical and complex designs, identification of the appropriate level for tests and full reporting of outcomes |
| ☐ | ☐ | Estimates of effect sizes (e.g. Cohen's *d*, Pearson's *r*), indicating how they were calculated |

*Our web collection on statistics for biologists contains articles on many of the points above.*

## Software and code

Policy information about availability of computer code

| Data collection | *Provide a description of all commercial, open source and custom code used to collect the data in this study, specifying the version used OR state that no software was used.* |
|---|---|
| Data analysis | *Provide a description of all commercial, open source and custom code used to analyse the data in this study, specifying the version used OR state that no software was used.* |

For manuscripts utilizing custom algorithms or software that are central to the research but not yet described in published literature, software must be made available to editors and reviewers. We strongly encourage code deposition in a community repository (e.g. GitHub). See the Nature Portfolio guidelines for submitting code & software for further information.

## Data

Policy information about availability of data

All manuscripts must include a data availability statement. This statement should provide the following information, where applicable:
- Accession codes, unique identifiers, or web links for publicly available datasets
- A description of any restrictions on data availability
- For clinical datasets or third party data, please ensure that the statement adheres to our policy

*Provide your data availability statement here.*

# Research involving human participants, their data, or biological material

Policy information about studies with human participants or human data. See also policy information about sex, gender (identity/presentation), and sexual orientation and race, ethnicity and racism.

| | |
|---|---|
| Reporting on sex and gender | *Use the terms sex (biological attribute) and gender (shaped by social and cultural circumstances) carefully in order to avoid confusing both terms. Indicate if findings apply to only one sex or gender; describe whether sex and gender were considered in study design; whether sex and/or gender was determined based on self-reporting or assigned and methods used. Provide in the source data disaggregated sex and gender data, where this information has been collected, and if consent has been obtained for sharing of individual-level data; provide overall numbers in this Reporting Summary. Please state if this information has not been collected. Report sex- and gender-based analyses where performed, justify reasons for lack of sex- and gender-based analysis.* |
| Reporting on race, ethnicity, or other socially relevant groupings | *Please specify the socially constructed or socially relevant categorization variable(s) used in your manuscript and explain why they were used. Please note that such variables should not be used as proxies for other socially constructed/relevant variables (for example, race or ethnicity should not be used as a proxy for socioeconomic status). Provide clear definitions of the relevant terms used, how they were provided (by the participants/respondents, the researchers, or third parties), and the method(s) used to classify people into the different categories (e.g. self-report, census or administrative data, social media data, etc.) Please provide details about how you controlled for confounding variables in your analyses.* |
| Population characteristics | *Describe the covariate-relevant population characteristics of the human research participants (e.g. age, genotypic information, past and current diagnosis and treatment categories). If you filled out the behavioural & social sciences study design questions and have nothing to add here, write "See above."* |
| Recruitment | *Describe how participants were recruited. Outline any potential self-selection bias or other biases that may be present and how these are likely to impact results.* |
| Ethics oversight | *Identify the organization(s) that approved the study protocol.* |

Note that full information on the approval of the study protocol must also be provided in the manuscript.

# Field-specific reporting

Please select the one below that is the best fit for your research. If you are not sure, read the appropriate sections before making your selection.

☐ Life sciences    ☐ Behavioural & social sciences    ☐ Ecological, evolutionary & environmental sciences

For a reference copy of the document with all sections, see nature.com/documents/nr-reporting-summary-flat.pdf

# Life sciences study design

All studies must disclose on these points even when the disclosure is negative.

| | |
|---|---|
| Sample size | *Describe how sample size was determined, detailing any statistical methods used to predetermine sample size OR if no sample-size calculation was performed, describe how sample sizes were chosen and provide a rationale for why these sample sizes are sufficient.* |
| Data exclusions | *Describe any data exclusions. If no data were excluded from the analyses, state so OR if data were excluded, describe the exclusions and the rationale behind them, indicating whether exclusion criteria were pre-established.* |
| Replication | *Describe the measures taken to verify the reproducibility of the experimental findings. If all attempts at replication were successful, confirm this OR if there are any findings that were not replicated or cannot be reproduced, note this and describe why.* |
| Randomization | *Describe how samples/organisms/participants were allocated into experimental groups. If allocation was not random, describe how covariates were controlled OR if this is not relevant to your study, explain why.* |
| Blinding | *Describe whether the investigators were blinded to group allocation during data collection and/or analysis. If blinding was not possible, describe why OR explain why blinding was not relevant to your study.* |

# Behavioural & social sciences study design

All studies must disclose on these points even when the disclosure is negative.

| | |
|---|---|
| Study description | *Briefly describe the study type including whether data are quantitative, qualitative, or mixed-methods (e.g. qualitative cross-sectional, quantitative experimental, mixed-methods case study).* |
| Research sample | *State the research sample (e.g. Harvard university undergraduates, villagers in rural India) and provide relevant demographic information (e.g. age, sex) and indicate whether the sample is representative. Provide a rationale for the study sample chosen. For studies involving existing datasets, please describe the dataset and source.* |

| | |
|---|---|
| Sampling strategy | *Describe the sampling procedure (e.g. random, snowball, stratified, convenience). Describe the statistical methods that were used to predetermine sample size OR if no sample-size calculation was performed, describe how sample sizes were chosen and provide a rationale for why these sample sizes are sufficient. For qualitative data, please indicate whether data saturation was considered, and what criteria were used to decide that no further sampling was needed.* |
| Data collection | *Provide details about the data collection procedure, including the instruments or devices used to record the data (e.g. pen and paper, computer, eye tracker, video or audio equipment) whether anyone was present besides the participant(s) and the researcher, and whether the researcher was blind to experimental condition and/or the study hypothesis during data collection.* |
| Timing | *Indicate the start and stop dates of data collection. If there is a gap between collection periods, state the dates for each sample cohort.* |
| Data exclusions | *If no data were excluded from the analyses, state so OR if data were excluded, provide the exact number of exclusions and the rationale behind them, indicating whether exclusion criteria were pre-established.* |
| Non-participation | *State how many participants dropped out/declined participation and the reason(s) given OR provide response rate OR state that no participants dropped out/declined participation.* |
| Randomization | *If participants were not allocated into experimental groups, state so OR describe how participants were allocated to groups, and if allocation was not random, describe how covariates were controlled.* |

# Ecological, evolutionary & environmental sciences study design

All studies must disclose on these points even when the disclosure is negative.

| | |
|---|---|
| Study description | *Briefly describe the study. For quantitative data include treatment factors and interactions, design structure (e.g. factorial, nested, hierarchical), nature and number of experimental units and replicates.* |
| Research sample | *Describe the research sample (e.g. a group of tagged Passer domesticus, all Stenocereus thurberi within Organ Pipe Cactus National Monument), and provide a rationale for the sample choice. When relevant, describe the organism taxa, source, sex, age range and any manipulations. State what population the sample is meant to represent when applicable. For studies involving existing datasets, describe the data and its source.* |
| Sampling strategy | *Note the sampling procedure. Describe the statistical methods that were used to predetermine sample size OR if no sample-size calculation was performed, describe how sample sizes were chosen and provide a rationale for why these sample sizes are sufficient.* |
| Data collection | *Describe the data collection procedure, including who recorded the data and how.* |
| Timing and spatial scale | *Indicate the start and stop dates of data collection, noting the frequency and periodicity of sampling and providing a rationale for these choices. If there is a gap between collection periods, state the dates for each sample cohort. Specify the spatial scale from which the data are taken* |
| Data exclusions | *If no data were excluded from the analyses, state so OR if data were excluded, describe the exclusions and the rationale behind them, indicating whether exclusion criteria were pre-established.* |
| Reproducibility | *Describe the measures taken to verify the reproducibility of experimental findings. For each experiment, note whether any attempts to repeat the experiment failed OR state that all attempts to repeat the experiment were successful.* |
| Randomization | *Describe how samples/organisms/participants were allocated into groups. If allocation was not random, describe how covariates were controlled. If this is not relevant to your study, explain why.* |
| Blinding | *Describe the extent of blinding used during data acquisition and analysis. If blinding was not possible, describe why OR explain why blinding was not relevant to your study.* |

Did the study involve field work? ☐ Yes ☐ No

# Field work, collection and transport

| | |
|---|---|
| Field conditions | *Describe the study conditions for field work, providing relevant parameters (e.g. temperature, rainfall).* |
| Location | *State the location of the sampling or experiment, providing relevant parameters (e.g. latitude and longitude, elevation, water depth).* |
| Access & import/export | *Describe the efforts you have made to access habitats and to collect and import/export your samples in a responsible manner and in compliance with local, national and international laws, noting any permits that were obtained (give the name of the issuing authority, the date of issue, and any identifying information).* |
| Disturbance | *Describe any disturbance caused by the study and how it was minimized.* |

# Reporting for specific materials, systems and methods

We require information from authors about some types of materials, experimental systems and methods used in many studies. Here, indicate whether each material, system or method listed is relevant to your study. If you are not sure if a list item applies to your research, read the appropriate section before selecting a response.

## Materials & experimental systems

| n/a | Involved in the study |
|---|---|
| ☐ ☐ | Antibodies |
| ☐ ☐ | Eukaryotic cell lines |
| ☐ ☐ | Palaeontology and archaeology |
| ☐ ☐ | Animals and other organisms |
| ☐ ☐ | Clinical data |
| ☐ ☐ | Dual use research of concern |
| ☐ ☐ | Plants |

## Methods

| n/a | Involved in the study |
|---|---|
| ☐ ☐ | ChIP-seq |
| ☐ ☐ | Flow cytometry |
| ☐ ☐ | MRI-based neuroimaging |

## Antibodies

Antibodies used — *Describe all antibodies used in the study; as applicable, provide supplier name, catalog number, clone name, and lot number.*

Validation — *Describe the validation of each primary antibody for the species and application, noting any validation statements on the manufacturer's website, relevant citations, antibody profiles in online databases, or data provided in the manuscript.*

## Eukaryotic cell lines

Policy information about cell lines and Sex and Gender in Research

Cell line source(s) — *State the source of each cell line used and the sex of all primary cell lines and cells derived from human participants or vertebrate models.*

Authentication — *Describe the authentication procedures for each cell line used OR declare that none of the cell lines used were authenticated.*

Mycoplasma contamination — *Confirm that all cell lines tested negative for mycoplasma contamination OR describe the results of the testing for mycoplasma contamination OR declare that the cell lines were not tested for mycoplasma contamination.*

Commonly misidentified lines (See ICLAC register) — *Name any commonly misidentified cell lines used in the study and provide a rationale for their use.*

## Palaeontology and Archaeology

Specimen provenance — *Provide provenance information for specimens and describe permits that were obtained for the work (including the name of the issuing authority, the date of issue, and any identifying information). Permits should encompass collection and, where applicable, export.*

Specimen deposition — *Indicate where the specimens have been deposited to permit free access by other researchers.*

Dating methods — *If new dates are provided, describe how they were obtained (e.g. collection, storage, sample pretreatment and measurement), where they were obtained (i.e. lab name), the calibration program and the protocol for quality assurance OR state that no new dates are provided.*

☐ Tick this box to confirm that the raw and calibrated dates are available in the paper or in Supplementary Information.

Ethics oversight — *Identify the organization(s) that approved or provided guidance on the study protocol, OR state that no ethical approval or guidance was required and explain why not.*

Note that full information on the approval of the study protocol must also be provided in the manuscript.

## Animals and other research organisms

Policy information about studies involving animals; ARRIVE guidelines recommended for reporting animal research, and Sex and Gender in Research

Laboratory animals — *For laboratory animals, report species, strain and age OR state that the study did not involve laboratory animals.*

| Wild animals | *Provide details on animals observed in or captured in the field; report species and age where possible. Describe how animals were caught and transported and what happened to captive animals after the study (if killed, explain why and describe method; if released, say where and when) OR state that the study did not involve wild animals.* |
|---|---|
| Reporting on sex | *Indicate if findings apply to only one sex; describe whether sex was considered in study design, methods used for assigning sex. Provide data disaggregated for sex where this information has been collected in the source data as appropriate; provide overall numbers in this Reporting Summary. Please state if this information has not been collected.  Report sex-based analyses where performed, justify reasons for lack of sex-based analysis.* |
| Field-collected samples | *For laboratory work with field-collected samples, describe all relevant parameters such as housing, maintenance, temperature, photoperiod and end-of-experiment protocol OR state that the study did not involve samples collected from the field.* |
| Ethics oversight | *Identify the organization(s) that approved or provided guidance on the study protocol, OR state that no ethical approval or guidance was required and explain why not.* |

Note that full information on the approval of the study protocol must also be provided in the manuscript.

## Clinical data

Policy information about clinical studies
All manuscripts should comply with the ICMJE guidelines for publication of clinical research and a completed CONSORT checklist must be included with all submissions.

| Clinical trial registration | *Provide the trial registration number from ClinicalTrials.gov or an equivalent agency.* |
|---|---|
| Study protocol | *Note where the full trial protocol can be accessed OR if not available, explain why.* |
| Data collection | *Describe the settings and locales of data collection, noting the time periods of recruitment and data collection.* |
| Outcomes | *Describe how you pre-defined primary and secondary outcome measures and how you assessed these measures.* |

## Dual use research of concern

Policy information about dual use research of concern

### Hazards

Could the accidental, deliberate or reckless misuse of agents or technologies generated in the work, or the application of information presented in the manuscript, pose a threat to:

No | Yes
- [ ] [ ] Public health
- [ ] [ ] National security
- [ ] [ ] Crops and/or livestock
- [ ] [ ] Ecosystems
- [ ] [ ] Any other significant area

### Experiments of concern

Does the work involve any of these experiments of concern:

No | Yes
- [ ] [ ] Demonstrate how to render a vaccine ineffective
- [ ] [ ] Confer resistance to therapeutically useful antibiotics or antiviral agents
- [ ] [ ] Enhance the virulence of a pathogen or render a nonpathogen virulent
- [ ] [ ] Increase transmissibility of a pathogen
- [ ] [ ] Alter the host range of a pathogen
- [ ] [ ] Enable evasion of diagnostic/detection modalities
- [ ] [ ] Enable the weaponization of a biological agent or toxin
- [ ] [ ] Any other potentially harmful combination of experiments and agents

# Plants

Seed stocks
*Report on the source of all seed stocks or other plant material used. If applicable, state the seed stock centre and catalogue number. If plant specimens were collected from the field, describe the collection location, date and sampling procedures.*

Novel plant genotypes
*Describe the methods by which all novel plant genotypes were produced. This includes those generated by transgenic approaches, gene editing, chemical/radiation-based mutagenesis and hybridization. For transgenic lines, describe the transformation method, the number of independent lines analyzed and the generation upon which experiments were performed. For gene-edited lines, describe the editor used, the endogenous sequence targeted for editing, the targeting guide RNA sequence (if applicable) and how the editor was applied.*

Authentication
*Describe any authentication procedures for each seed stock used or novel genotype generated. Describe any experiments used to assess the effect of a mutation and, where applicable, how potential secondary effects (e.g. second site T-DNA insertions, mosiacism, off-target gene editing) were examined.*

# ChIP-seq

## Data deposition

☐ Confirm that both raw and final processed data have been deposited in a public database such as GEO.

☐ Confirm that you have deposited or provided access to graph files (e.g. BED files) for the called peaks.

Data access links
*May remain private before publication.*
*For "Initial submission" or "Revised version" documents, provide reviewer access links.  For your "Final submission" document, provide a link to the deposited data.*

Files in database submission
*Provide a list of all files available in the database submission.*

Genome browser session
(e.g. UCSC)
*Provide a link to an anonymized genome browser session for "Initial submission" and "Revised version" documents only, to enable peer review.  Write "no longer applicable" for "Final submission" documents.*

## Methodology

Replicates
*Describe the experimental replicates, specifying number, type and replicate agreement.*

Sequencing depth
*Describe the sequencing depth for each experiment, providing the total number of reads, uniquely mapped reads, length of reads and whether they were paired- or single-end.*

Antibodies
*Describe the antibodies used for the ChIP-seq experiments; as applicable, provide supplier name, catalog number, clone name, and lot number.*

Peak calling parameters
*Specify the command line program and parameters used for read mapping and peak calling, including the ChIP, control and index files used.*

Data quality
*Describe the methods used to ensure data quality in full detail, including how many peaks are at FDR 5% and above 5-fold enrichment.*

Software
*Describe the software used to collect and analyze the ChIP-seq data. For custom code that has been deposited into a community repository, provide accession details.*

# Flow Cytometry

## Plots

Confirm that:

☐ The axis labels state the marker and fluorochrome used (e.g. CD4-FITC).

☐ The axis scales are clearly visible. Include numbers along axes only for bottom left plot of group (a 'group' is an analysis of identical markers).

☐ All plots are contour plots with outliers or pseudocolor plots.

☐ A numerical value for number of cells or percentage (with statistics) is provided.

## Methodology

Sample preparation
*Describe the sample preparation, detailing the biological source of the cells and any tissue processing steps used.*

Instrument
*Identify the instrument used for data collection, specifying make and model number.*

Software
*Describe the software used to collect and analyze the flow cytometry data. For custom code that has been deposited into a community repository, provide accession details.*

| Cell population abundance | *Describe the abundance of the relevant cell populations within post-sort fractions, providing details on the purity of the samples and how it was determined.* |
| Gating strategy | *Describe the gating strategy used for all relevant experiments, specifying the preliminary FSC/SSC gates of the starting cell population, indicating where boundaries between "positive" and "negative" staining cell populations are defined.* |

☐ Tick this box to confirm that a figure exemplifying the gating strategy is provided in the Supplementary Information.

# Magnetic resonance imaging

## Experimental design

| Design type | *Indicate task or resting state; event-related or block design.* |
| Design specifications | *Specify the number of blocks, trials or experimental units per session and/or subject, and specify the length of each trial or block (if trials are blocked) and interval between trials.* |
| Behavioral performance measures | *State number and/or type of variables recorded (e.g. correct button press, response time) and what statistics were used to establish that the subjects were performing the task as expected (e.g. mean, range, and/or standard deviation across subjects).* |

## Acquisition

| Imaging type(s) | *Specify: functional, structural, diffusion, perfusion.* |
| Field strength | *Specify in Tesla* |
| Sequence & imaging parameters | *Specify the pulse sequence type (gradient echo, spin echo, etc.), imaging type (EPI, spiral, etc.), field of view, matrix size, slice thickness, orientation and TE/TR/flip angle.* |
| Area of acquisition | *State whether a whole brain scan was used OR define the area of acquisition, describing how the region was determined.* |

Diffusion MRI   ☐ Used   ☐ Not used

## Preprocessing

| Preprocessing software | *Provide detail on software version and revision number and on specific parameters (model/functions, brain extraction, segmentation, smoothing kernel size, etc.).* |
| Normalization | *If data were normalized/standardized, describe the approach(es): specify linear or non-linear and define image types used for transformation OR indicate that data were not normalized and explain rationale for lack of normalization.* |
| Normalization template | *Describe the template used for normalization/transformation, specifying subject space or group standardized space (e.g. original Talairach, MNI305, ICBM152) OR indicate that the data were not normalized.* |
| Noise and artifact removal | *Describe your procedure(s) for artifact and structured noise removal, specifying motion parameters, tissue signals and physiological signals (heart rate, respiration).* |
| Volume censoring | *Define your software and/or method and criteria for volume censoring, and state the extent of such censoring.* |

## Statistical modeling & inference

| Model type and settings | *Specify type (mass univariate, multivariate, RSA, predictive, etc.) and describe essential details of the model at the first and second levels (e.g. fixed, random or mixed effects; drift or auto-correlation).* |
| Effect(s) tested | *Define precise effect in terms of the task or stimulus conditions instead of psychological concepts and indicate whether ANOVA or factorial designs were used.* |

Specify type of analysis:   ☐ Whole brain   ☐ ROI-based   ☐ Both

| Statistic type for inference<br>(See Eklund et al. 2016) | *Specify voxel-wise or cluster-wise and report all relevant parameters for cluster-wise methods.* |

| Correction | *Describe the type of correction and how it is obtained for multiple comparisons (e.g. FWE, FDR, permutation or Monte Carlo).* |

## Models & analysis

| n/a | Involved in the study |
|:---:|:---|
| ☐ | ☐ Functional and/or effective connectivity |
| ☐ | ☐ Graph analysis |
| ☐ | ☐ Multivariate modeling or predictive analysis |

**Functional and/or effective connectivity**

*Report the measures of dependence used and the model details (e.g. Pearson correlation, partial correlation, mutual information).*

**Graph analysis**

*Report the dependent variable and connectivity measure, specifying weighted graph or binarized graph, subject- or group-level, and the global and/or node summaries used (e.g. clustering coefficient, efficiency, etc.).*

**Multivariate modeling and predictive analysis**

*Specify independent variables, features extraction and dimension reduction, model, training and evaluation metrics.*

