## [Peer Review File · Nature Structural & Molecular Biology]

tRNA as an assembly chaperone for a macromolecular transcription-processing complex

Corresponding Author: Dr Clemens Grimm

Version 0:

Decision Letter:

5th Dec 2024

Dear Dr. Grimm,

Thank you again for submitting your manuscript "tRNA as an assembly chaperone for a macromolecular transcription-processing complex". I'm writing to let you know that we have decided to send your manuscript for peer review.

I am re-opening the manuscript submission link for you to resubmit your manuscript with all the associated files needed for the peer review process directly to our system, at your convenience. Please see below for details regarding the required materials. Please follow the link at the bottom of this email to upload the documents.

We want to ensure that the methods and statistics reporting in our papers are of the highest quality. To that end, we ask authors to fill out a Reporting Summary that collects information on experimental design and reagents, as well as an editorial Policy Checklist, which confirms compliance with our editorial policies, including the declaration of Competing Interests. If your paper includes ChIP-seq, flow cytometry or MRI data, we ask you take special care to complete those sections of the Reporting Summary as this data will aid greatly in the review of your manuscript.

These documents can be found by following the links below:

Reporting Summary:
<https://www.nature.com/documents/nr-reporting-summary.pdf>

Editorial Policy Checklist: <https://www.nature.com/documents/nr-editorial-policy-checklist.pdf>

Please be aware of our guidelines on digital image standards.

IMPORTANT

In order for us to proceed with the peer review process of your manuscript, we require you to provide accession numbers and reviewer tokens to access sequencing data sets. Please add this information to your manuscript file.

Please note we require official wwPDB validation reports for newly described atomic structures, as noted in the policy checklist. We also request that authors provide cryo-EM maps, half-maps and models, to help the reviewers in assessing the work. We recommend the use of figshare integration into our systems, which allows for provision of anonymous access links for the referees (<https://www.springernature.com/gp/authors/research-data/figshare-integration>).

Alternatively, please upload .zip folders directly with the submission. To ensure the ease of reviewer access to the data, please specify in the Data Availability section, where the files can be found (provide a figshare link or direct the reader to the manuscript files).

Additionally, if your manuscript contains custom code, I would like to kindly request that you provide the code used to analyse the data to the reviewers. In order for the reviewers to evaluate the work adequately they must be able to test the software/review the code themselves. If you have not yet provided the software, we therefore request that you provide a single compressed zip file containing the software with a readme.txt file or other user manual containing complete instructions for installing and running the software. If appropriate, please also provide example data and expected output.

Sufficient material should be provided for referees to directly test the performance of the software/algorithm. If the software and materials are small enough to fit in a single compressed zip file less than 6MB in size, you may email this file directly to me. If the zip file is between 6 MB and 200 MB you may upload it to our file transfer site. If necessary, a second zip file up to 200 MB in size can be used to supply the example data. Please let me know if you need to use this option and I'll send you further details. Alternatively, you can also upload the code to GitHub and provide us with the link.

Please also fill out and return to me the code and software submission checklist that will be made available to editors and reviewers during manuscript assessment. Please note that this form is a dynamic 'smart pdf' and must therefore be downloaded and completed in Adobe Reader, instead of opening it in a web browser.

<https://www.nature.com/documents/nr-software-policy.pdf>

Please use the link below to submit the files. **Please also remember to move forward all other manuscript files associated with this version of the paper.**

Link Redacted

Sincerely,
Sara

Sara Osman, Ph.D.
Senior Editor
Nature Structural & Molecular Biology

Version 1:

Decision Letter:

8th Jan 2025

Dear Dr. Grimm,

Thank you again for submitting your manuscript "tRNA as an assembly chaperone for a macromolecular transcription-processing complex". We now have comments (below) from the 3 reviewers who evaluated your paper. In light of those reports, we remain interested in your study and would like to see your response to the comments of the referees, in the form of a revised manuscript.

You will see that while all three reviewers appreciate the insights provided by this work, Reviewer #1 requests clarifying the use of nomenclature throughout the manuscript, asks for more detailed figure legends and method descriptions, and requests performing in vitro binding assays with mutated tRNA-Gln to test the induced-fit hypothesis. Reviewer #2, requests further discussion of the biological significance of tRNA lacking mcm5s2U being enriched in vRNAP, and the possible impact on translation of the sequestration of tRNAGln and tRNAArg during vRNAP formation and release during PIC assembly, and the criteria for the selection of these tRNAs. Additionally, they request improvement of some of the models, and more detailed description of observed structural features, better quantification of tRNA modification rates, and comparison of the binding efficiencies of intermediates to substantiate claims about the role of tRNA in association of D1/D12 with vRNAP. Furthermore, Reviewer #3 suggests purifying untagged RNAP from virion cores to confirm which tRNAs are bound, and tone down some of the speculations in the discussion. Please be sure to address/respond to all concerns of the referees in full in a point-by-point response and highlight all changes in the revised manuscript text file. If you have comments that are intended for editors only, please include those in a separate cover letter.

We expect to see your revised manuscript within 3 months. If you cannot send it within this time, please contact us to discuss an extension; we would still consider your revision, provided that no similar work has been accepted for publication at NSMB or published elsewhere.

Reporting Summary:

EXTENDED DATA FIGURES

Please note that all key data shown in the main figures as cropped gels or blots should be presented in uncropped form, with molecular weight markers. These data can be aggregated into a single supplementary figure item. While these data can be displayed in a relatively informal style, they must refer back to the relevant figures. These data should be submitted with the final revision, as source data, prior to acceptance, but you may want to start putting it together at this point.

SOURCE DATA: we request that authors provide, in tabular form, the data underlying the graphical representations used in figures. This is to further increase transparency in data reporting, as detailed in this editorial (<http://www.nature.com/nsmb/journal/v22/n10/full/nsmb.3110.html>). Spreadsheets can be submitted in excel format. Only one (1) file per figure is permitted; thus, for multi-paneled figures, the source data for each panel should be clearly labeled in the Excel file; alternately the data can be provided as multiple, clearly labeled sheets in an Excel file. When submitting files, the title field should indicate which figure the source data pertains to. We encourage our authors to provide source data at the revision stage, so that they are part of the peer-review process.

Data availability: this journal strongly supports public availability of data. All data used in accepted papers should be available via a public data repository, or alternatively, as Supplementary Information. If data can only be shared on request, please explain why in your Data Availability Statement, and also in the correspondence with your editor. Please note that for some data types, deposition in a public repository is mandatory - more information on our data deposition policies and available repositories can be found below:

<https://www.nature.com/nature-research/editorial-policies/reporting-standards#availability-of-data>

Link Redacted

Sincerely,
Sara

Sara Osman, Ph.D.
Senior Editor
Nature Structural & Molecular Biology

Referee expertise:

Referee #1: tRNA processing, cryo-EM

Referee #2: tRNA mechanisms, structural biology

Referee #3: Poxviruses, RNA processing

Reviewers' Comments:

Reviewer #1 (Remarks to the Author):

In this manuscript Bartuli et al characterize investigate the role of tRNA in the formation of the poxviral RNA polymerase complex during poxviral infection. Previous work established that tRNA is part of the viral RNA polymerase complex but its function in the complex remained unknown. The authors used cryo-EM to capture a series of assembly intermediates, which revealed a complex mechanism of assembly. Careful analysis of the high resolution complete vRNAP structure revealed that the tRNA is bound in an unusual, strained conformation. Comparison with other tRNAs revealed that this unusual conformation can only be formed by certain tRNA-Gln and tRNA-Arg anticodons, explaining vRNAP's specificity. The authors hypothesized that tRNA modifications may play a role in vRNAP formation. To test this hypothesis, they tested binding of unmodified in vitro transcribed tRNA-Gln and found that it is less effective at promoting vRNAP assembly. Then they used mass-spec to determine that several tRNA modifications are present in the vRNAP associated tRNAs, however it was noted that the mcm5-s2-U modification, which occurs at position U34 was not detected. Modeling of this modification onto the structure suggests it would cause a steric clash. Finally, the authors used proteomic approaches to look at the RNA and proteins present in purified vaccinia virions. 60% of the RNA was found to be tRNA-Gln and tRNA-Arg supporting the structural observations in the manuscript.

Overall, the authors present compelling structural and functional data which reveals significant new insight into the role that the tRNA-Gln and tRNA-Arg play in chaperoning assembly of the vRNAP following poxviral infection. I am supportive of publication once the authors address the following concerns.

- The use of tRNAGln/Arg nomenclature throughout the manuscript is confusing. I assume the authors use this to mean either tRNAGln or tRNAArg as both can function as a chaperone but not at the same time. I recommend replacing this with a better acronym such as "heterogenous tRNAGln/Arg". Moreover, based on the methods my assumption is that the tRNA used in all cryo-EM structures was a heterogenous mixture of the two tRNAs. If this is the case then the authors need to make this very clear in the results, figures/legends, and deposited PDBs. Figure 2 and Figure 5 for example use the label tRNAGln on the structures which is misleading.
- The authors should provide more details on how the in vitro reconstitution experiments were performed. The tRNA was first purified with vRNAP and then extracted, labeled, and re-incubated with vRNAP?? I recommend moving Extended Data Fig. 1A to Fig. 1 to help readers better understand how these reconstitution experiments were performed. Using colored labels for all of the factors/proteins would also help readers better connect Figure 1 with Figures 2/3. For example, it is hard to understand which factors make up the minimal vs Core vs Complete vRNAP.
- Figure 2 – Please add more details to the figure legend to help readers understand this complicated figure.
- Figure 3 – It is difficult to understand which structures in this figure were already determined and which ones are new to this manuscript. The authors need to include the PDB codes for all published manuscripts in the figure and legend.
- The use of different names for the intermediates was hard to keep track of. In particular the authors use both I6 and "complete vRNAP" to refer to the same state. It would be better to pick one name and use that throughout the manuscript.

- The authors should add a supplemental figure/panel showing the full EM density for the tRNA in the complete vRNAP structure, with one tRNA-Gln and one tRNA-Arg individually docked into the density. Aside from the noted rearrangements in the ASL are there any other distinguishing features of these tRNAs? What about the different codons and isodecoders of tRNA Gln and tRNA Arg? Perhaps a supplemental table showing the sequence of all these tRNAs, with the specificity elements and putative modification sites indicated would be helpful.
- Figure 5 – In panel (A) or (D) the authors should include an overlay of the cryo-EM density. (F) is not defined in the figure legend. (H) Please add the other tRNA Gln anticodon (CUG). Inspection of the tRNA database did not reveal a high confidence human tRNA-Arg-GCG so perhaps it is worth mentioning why this one is not included.
- To test their induced-fit hypothesis the authors should perform in-vitro binding assays with transcribed tRNA-Gln containing mutations at position 34 and 35.
- Figure 8 – Can quantitative MS distinguish the different populations of tRNA-Gln and tRNA-Arg isodecoders present? If so, it would be interesting to provide this information as a supplemental table. Are the ratios of tRNA-Gln vs tRNA-Arg reflective of cellular levels of these tRNAs or does vRNAP have a slight preference for tRNA-Gln?
- Have the authors attempted to determine EM structures with in vitro transcribed tRNAs to avoid issues with heterogeneity in their samples?

Reviewer #2 (Remarks to the Author):

This manuscript serves as a follow-up to the papers reported by this group in 2019 and 2021, providing a detailed role of tRNA in the assembly process of vRNAP. The elucidation of the function of tRNA as an assembly chaperone for vRNAP is particularly intriguing. This reviewer strongly supports its publication, provided the following comments that should be addressed:

The finding that tRNA lacking mcm5s2U is enriched in vRNAP is fascinating, but its biological significance requires further discussion. Fully-modified tRNAs are efficiently aminoacylated and likely sequestered by EF1 alpha. If tRNA^{Gln} and tRNA^{Arg} lacking mcm5s2U are less aminoacylated, they might avoid competition with EF1 alpha and be preferentially utilized for vRNAP assembly. This might be one of advantage of utilization of hypomodified tRNA as an assembly factor.

Hypomodified tRNA^{Gln} and tRNA^{Arg} are sequestered during vRNAP formation and later released during PIC assembly. Does this process have any impact on translation? In particular, codon optimality might be changed. A discussion from this perspective would be valuable.

The manuscript discusses three limitations to explain the selection of tRNA^{Gln} and tRNA^{Arg}. Are these criteria sufficient? Could other characteristics of tRNA play a role in this selection?

The distorted structural changes in the ASL within the N-lobe of NPH-I are intriguing, but the description of Fig. 5F is insufficient. A more detailed explanation of the interactions between U35, A37, and the amino acid residues in the pocket, as well as the positioning of Mg²⁺ ions, is needed.

The ASL model of tRNA in Fig. 5BC requires improvement. In this model, A37 and U33 are paired; however, in a free form of tRNA, the anticodon bases and positions 37 and 38 are stacked, while U33 typically forms a U-turn structure.

The claim that mcm5s2U is low in uninfected cells and increases during infection in the nucleoside analysis, as illustrated in Fig. 6CD. However, there is no information about normalization of the peak intensity in the experiment. More importantly, mcm5s2U nucleoside level can be changed by the abundance of four tRNA species carrying mcm5s2U. This data does not provide any information about tRNA modification status without quantifying the tRNA abundance. To quantify tRNA modification rates, it would be necessary to isolate and purify tRNA^{Gln} and tRNA^{Arg} and analyze the anticodon-containing fragments to measure the ratio of modified to unmodified fragments. Additionally, intermediates such as s2U34 and mcm5U34, which precede mcm5s2U34, should also be verified. cm5s2U is a precursor to mcm5s2U but is known to be converted to ncm5s2U and is scarcely present in cells. The detection of cm5s2U is likely due to hydrolysis of the terminal methyl ester during tRNA preparation.

The data in Fig. 1B cannot be interpreted as showing that tRNA binding promotes the association of D1/D12 with vRNAP. Rather, the data suggest that the binding of D1/D12 to vRNAP further stabilizes tRNA binding. A comparison of the binding efficiency of D1/D12 with the I3 and I1 intermediates is necessary.

Minor points:

CE(D1/D12) should be written as D1/D12 in text, because this heterodimer is shown as D1 and D12 separately in many Figures.

If authors use previous structural data (structure I1, PIC, relaxed tRNA model, etc.), the PDB ID should be shown for these structures in text, Figure legend and Data availability sections.

Typo in Fig3 legend lane 7; I5: complete vRNAP → I6: complete vRNAP.

In Figure 5, the legend of (F) is lacking.

The number of methylation position should be shown in uppercase. (ex. m1A, m5C)

The model of U34 in Fig7C is actually s2U. Please correct it.

Extended Fig1: B, the arrow indicating NPH-I degradation product is not shown. You should label the sample name of each lane.

C, You should mention and discuss the weak binding between D1/D12 and NPH-I/E11 in main text.

D, You should label the sample name of each gel in Figure.

Extended Fig2: B, I'm interested in the difference among Class 1 to 3. Please address it in figure legend or main text. Final resolution of 3.3A is not consistent with FSC curve data (Ex Fig2B).

E, The letter '3.8A' to the left of the letter 'E' is probably a mistake.

Extended Fig4: B, The map of class 2 from 1st Classification is slightly strange. (The density at the top is distorted.) You should mention about class 1 from 2nd classification in main text. I'm also interested in remaining particles (about 240K particles) in 3rd classification. Is it only junk, or other conformations?

Extended Fig5; You should clearly label 'I5' in Figure or Figure legends.

Extended Data Table1; You should clearly label 'Intermediate 5' to 8C8H column.

You should write the number of ions in the same way in all column.

Particles (automatically selected) of Intermediate 3 and 4 are mistaken.

Number of movies of Intermediate 3 is mistaken.

P46 lane4 Fig1C→ Fig1B

P46 lane11 the amounts of E11 (90 nmol) is correct?

P47 lane 7 I can't find total calf tRNA of Roche using this number.

P47 You need to show the purification protocol for recombinant E11.

P49 You should address the details of purification protocol of vRNAP samples in separate section. I can't find the details of the preparation of Intermediate 5 sample.

Reviewer #3 (Remarks to the Author):

A previous landmark study (Cell, 2019) by Prof. Fischer and colleagues showed that the transcriptionally active “complete” vaccinia virus RNA polymerase complex comprised 15 viral polypeptides and uncharged host tRNA-Gln (and to lesser extent host tRNA-Arg). tRNA-Gln interacts with Rap94 (required for early gene transcription initiation), VETF (early promoter binding initiation factor) and NPH-I (a DExH-box transcription termination factor). They proposed that tRNA is important for stability of the vRNAP complex.

Here, Bartuli et al. test their hypothesis that tRNA serves an assembly chaperone for vRNAP. Starting with an 8-subunit minimal RNAP (lacking Rap94) and “core” RNAP (containing Rap94), they sought to reconstitute complete vRNAP by adding back purified components. They find that NPH-I and E11 suffice for recruitment of 3' 32P-labeled tRNA into the core RNAP complex, generating an assembly intermediate that was capable of recruiting the vaccinia capping enzyme.

The authors then proceed to solve cryo-EM structures (2.5 to 3.9 Å resolution) of assembly intermediates and complete vRNAPs (+VETF, lacking capping enzyme or –VETF, +capping enzyme) that provide additional insights into the interactions of the tRNA with protein components and protein-protein interactions.

Focusing on the structure of the vRNAP-associated tRNA-Gln, versus a model of the native tRNA fold, they find that the wobble base U34 remains flipped out whereas the rest of the anticodon stem-loop adopts a different fold that they term “strained,” entailing rearranged nucleobase hydrogen bonds. They make an argument that tRNA-Gln and tRNA-Arg are uniquely capable of adopting this strained anticodon stem-loops conformation.

A valuable component of the study is the inclusion of an LC–MS analysis of the modification status of the nucleosides of vRNAP-associated tRNA-Gln, which revealed that the RNAP-bound tRNA-Gln has the expected ribose and base methylations but lacks the mcm5s2U wobble modification. They affirm this by showing that the RNAP-bound tRNA-Gln resists anticodon incision by γ -toxin (which cleaves at mcm5s2U).

Finally, the authors document enrichment of tRNA-Gln/Arg in infectious vaccinia and Mpox virions. Albeit they do not demonstrate that the tRNAs are associated directly with the virion RNAP. tRNA-Gly is nearly as abundant in vaccinia virions as tRNA-Arg. What is tRNA-Gly doing in the vaccinia virus particle – and is it associated with virion RNAP? Is tRNA-Gly also enriched in Mpox virions? Inclusion of a probe for tRNA-Gly in Fig. 8C would be useful. It is relatively easy to purify RNAP from virion cores (without any tags), so perhaps the authors would consider doing this to confirm which tRNAs are bound to RNAP in virions.

The Discussion provides a concise summary of key points. I think the speculation that local depletion of Gln and Arg amino acids at viral replication sites during the late stage of vaccinia infection prompts the development of an uncharged, wobble-unmodified tRNA-Gln/Arg pool for association with vRNAP is a stretch, especially given that translation of viral late mRNAs remains quite vigorous at late times when assembly of new virions occurs. (Moreover, the cell culture medium (DMEM) for vaccinia virus growth is typically supplemented with glutamine, so it is not clear that Gln is limiting during virus infection.) Perhaps the authors might temper their speculations at end of Discussion.

Overall, this is an important study, rigorously performed, that sheds light on the assembly and structure of the poxvirus RNAP responsible for early gene transcription. The reliance of a specific tRNA and a strained tRNA conformation (as well as lack of a base modification) to allow for tRNA function as an assembly chaperone has broad implications for an expanding view of tRNA's biological functions.

Version 2:

Decision Letter:

Our ref: NSMB-A50194B

9th Apr 2025

Dear Dr. Grimm,

Thank you for submitting your revised manuscript "tRNA as an assembly chaperone for a macromolecular transcription-processing complex" (NSMB-A50194B). It has now been seen by the original referees and their comments are below. The reviewers find that the paper has improved in revision, and therefore we'll be happy in principle to publish it in Nature Structural & Molecular Biology, pending minor revisions to satisfy the referees' final requests and to comply with our editorial and formatting guidelines.

We are now performing detailed checks on your paper and will send you a checklist detailing our editorial and formatting requirements in the next few weeks. Please do not upload the final materials and make any revisions until you receive this additional information from us.

To facilitate our work at this stage, it is important that we have a copy of the main text as a word file. If you could please send along a word version of this file as soon as possible, we would greatly appreciate it; please make sure to copy the NSMB account (cc'ed above).

Sincerely,
Sara

Sara Osman, Ph.D.
Senior Editor
Nature Structural & Molecular Biology

Reviewer #1 (Remarks to the Author):

The authors have done an excellent job addressing my previous concerns and those of the other reviewers. I appreciate the updated figures and clarification on the heterogeneity of the tRNA in their structures. I am fully supportive of publication. Congratulations to the authors on this elegant mechanistic work revealing how tRNA-Gln and tRNA-Arg chaperone assembly of the vRNAP following poxviral infection.

Minor Note:

The size of several panels in Extended Data Fig. 4 should be increased.

Reviewer #2 (Remarks to the Author):

The authors adequately addressed this reviewer's comment. I still find some typo given below.

Tryzol > TRIzol
Invitrogene > Invitrogen
Acrylamid > Acrylamide
Nothern blot > Northern blot
There is a numbering issue in the legend of Figure 5B-E.

Reviewer #3 (Remarks to the Author):

The revisions are satisfactory

Version 3:

Decision Letter:

14th Jul 2025

Dear Dr. Grimm,

We are now happy to accept your revised paper "tRNA as an assembly chaperone for a macromolecular transcription-processing complex" for publication as an article in Nature Structural & Molecular Biology.

Your paper will be published online soon after we receive proof corrections and will appear in print in the next available issue. You can find out your date of online publication by contacting the production team shortly after sending your proof corrections.

An online order form for reprints of your paper is available at <https://www.nature.com/reprints/author-reprints.html>. Please let your coauthors and your institutions'

public affairs office know that they are also welcome to order reprints by this method.

Sincerely,
Sara

Sara Osman, Ph.D.
Senior Editor
Nature Structural & Molecular Biology

We thank the reviewers for their helpful comments on our manuscript and have responded to them as outlined below in our point-by-point response (reviewer's comments are labelled in black italic, our response in blue letters):

Reviewer #1 (Remarks to the Author):

• The use of heterogenous tRNA^{Gln}/Arg nomenclature throughout the manuscript is confusing. I assume the authors use this to mean either heterogenous tRNA^{Gln} or heterogenous tRNA^{Arg} as both can function as a chaperone but not at the same time. I recommend replacing this with a better acronym such as “heterogenous tRNA^{Gln/Arg}”. Moreover, based on the methods my assumption is that the tRNA used in all cryo-EM structures was a heterogenous mixture of the two tRNAs. If this is the case then the authors need to make this very clear in the results, figures/legends, and deposited PDBs. Figure 2 and Figure 5 for example use the label tRNA^{Gln} on the structures which is misleading.

We thank the referee for bringing up this important point and agree that the labelling of the tRNAs was misleading at some positions in the manuscript. We now state at the first mentioning of the vRNAP-associated tRNAs that they are collectively termed tRNA^{Gln/Arg} throughout this manuscript (see page 2). Only in Figure 5 and 7, we specifically focus on tRNA^{Gln} as coordinates in the complete vRNAP model as deposited in the PDB. In this model, the density was built with the dominant tRNA^{Gln} species only.

In addition, we now present also a new Extended Data Fig. 7. Here we compare the tRNA^{Gln}-UUG model with its associated cryo EM density as extracted from the complete vRNAP structure to a model for the tRNA^{Arg}-CCG sequence, modelled manually into the complete vRNAP density. While both sequences would fit the density reasonably, the tRNA^{Gln} sequence fit is better. This is expected since our experimental data show that tRNA^{Gln} is the dominating species in complete vRNAP.

• The authors should provide more details on how the in vitro reconstitution experiments were performed. The tRNA was first purified with vRNAP and then extracted, labeled, and re-incubated with vRNAP?? I recommend moving Extended Data Fig. 1A to Fig. 1 to help readers better understand how these reconstitution experiments were performed. Using colored labels for all of the factors/proteins would also help readers better connect Figure 1 with Figures 2/3. For example, it is hard to understand which factors make up the minimal vs Core vs Complete vRNAP.

We have now revised Fig. 1 to provide a clearer overview of our in vitro reconstitution experiments. As suggested by the reviewer, we have moved the relevant panel from Extended Data Fig. 1A into the main Fig. 1 to illustrate in a scheme how the reconstitution of complete vRNAP was performed. In our revised Figures, we have now also used a color code for labelling the different factors investigated in this study. This very same color code is also used in the structural analysis of the manuscript.

• *Figure 2 – Please add more details to the figure legend to help readers understand this complicated figure.*

As requested by this reviewer, we have added additional descriptions and comments to the figure legend to make this figure better understandable.

• *Figure 3 – It is difficult to understand which structures in this figure were already determined and which ones are new to this manuscript. The authors need to include the PDB codes for all published manuscripts in the figure and legend.*

We agree with the referee and have adapted Figure 3 and its legend accordingly to illustrate, which structures had been reported previously and which are newly presented in this manuscript.

• *The use of different names for the intermediates was hard to keep track of. In particular the authors use both I6 and “complete vRNAP” to refer to the same state. It would be better to pick one name and use that throughout the manuscript.*

We agree with the referee and have modified figures, figure legends and the manuscript text accordingly. Specifically, we now replaced the term “I6” with the term “complete vRNAP” throughout the text.

• *The authors should add a supplemental figure/panel showing the full EM density for the tRNA in the complete vRNAP structure, with one tRNA-Gln and one tRNA-Arg individually docked into the density. Aside from the noted rearrangements in the ASL are there any other distinguishing features of these tRNAs? What about the different codons and isodecoders of tRNA-Gln and tRNA Arg? Perhaps a supplemental table showing the sequence of all these tRNAs, with the specificity elements and putative modification sites indicated would be helpful.*

We agree with the referee and have added an additional Figure (Extended Data Fig. 7), which shows the cryo-EM density of the tRNA fitted with models for the Gln-UUG-1-1 and the Arg-CCG-1-1 sequences. We have also added an alignment of the relevant isodecoder sequences as extended data as a supplemental table. In addition, we show in a new Fig. 5I stem/loop sequences of all relevant isodecoders, confirming the specificity in the tRNA selection mechanism.

• *Figure 5 – In panel (A) or (D) the authors should include an overlay of the cryo-EM density.*

We thank the referee for this valuable suggestion. We have added a new panel A to Fig. 5, which now shows the anticodon region superposed by a cryo-EM density isosurface.

(F) is not defined in the figure legend.

Corrected. We thank the referee for bringing up this mistake. Please also note that there are comprehensive changes to all panels and their letters due to the inclusion of new images/data.

(H) Please add the other tRNA Gln anticodon (CUG).

We have added the Gln-CUG sequence as suggested by the reviewer.

Inspection of the tRNA database did not reveal a high confidence human tRNA-Arg-GCG so perhaps it is worth mentioning why this one is not included.

We thank the reviewer for the valuable information that there is no high confidence tRNA-Arg-GCG sequence in the database. We have added this information in the legend to Fig. 5I (Fig. 5H in the original version of the manuscript).

• To test their induced-fit hypothesis the authors should perform in-vitro binding assays with transcribed tRNA-Gln containing mutations at position 34 and 35.

We thank the reviewer for this valuable suggestion. We performed a binding/bandshift assay and show its result in the new Extended Data Fig. 8. Whereas tRNA^{Gln/Arg} isolated from complete vRNAP induces assembly with high efficiency, in vitro transcribed tRNA^{Gln} is less efficient, thus confirming the results shown in Fig. 6A. In contrast, a single mutation in the anticodon stem/loop region, substituting the critical residue U33 to G abolishes its reconstitution activity. These results not only confirm the proposed induced-fit binding and selection mechanism but also illustrate the importance of base modifications in the selection mechanism.

• Figure 8 – Can quantitative MS distinguish the different populations of tRNA-Gln and tRNA-Arg isodecoders present? If so, it would be interesting to provide this information as a supplemental table. Are the ratios of tRNA-Gln vs tRNA-Arg reflective of cellular levels of these tRNAs or does vRNAP have a slight preference for tRNA-Gln?

Unfortunately, the experimental setup of our MS analysis does not allow us to determine this interesting aspect since it is based on a total digest of isolated tRNA. So far, we have no indication that the ratio of the two tRNA species reflect their cellular levels. We would like to point out, however, that the amount of tRNA absorbed by complete vRNAP is very low as compared to the total tRNA pool in the infected cell.

• Have the authors attempted to determine EM structures with in vitro transcribed tRNAs to avoid issues with heterogeneity in their samples?

We have tried these experiments but failed to obtain stable particles amenable for cryo-EM analysis. As can be seen from the binding assays shown in Fig. 6A and the new Extended Data

Fig. 8, the efficiency of assembly is very low when in vitro transcribed tRNAs are used. This makes an in vitro reconstitution and structural analysis of vRNAP assembly intermediates technically impossible. It further confirms our findings that modifications in the tRNA play a major role in complex stability and assembly activity.

Reviewer #2 (Remarks to the Author):

This manuscript serves as a follow-up to the papers reported by this group in 2019 and 2021, providing a detailed role of tRNA in the assembly process of vRNAP. The elucidation of the function of tRNA as an assembly chaperone for vRNAP is particularly intriguing. This reviewer strongly supports its publication, provided the following comments that should be addressed:

The finding that tRNA lacking mcm5s2U is enriched in vRNAP is fascinating, but its biological significance requires further discussion. Fully-modified tRNAs are efficiently aminoacylated and likely sequestered by EF1alpha. If tRNA^{Gln} and tRNA^{Arg} lacking mcm5s2U are less aminoacylated, they might avoid competition with EF1alpha and be preferentially utilized for vRNAP assembly. This might be one of advantage of utilization of hypomodified tRNA as an assembly factor.

We thank the reviewer for this interesting aspect and have now modified the discussion as suggested:

“In uninfected cells, tRNAs are predominantly associated with cellular factors such as their cognate aminoacyl-tRNA synthetases and elongation factors (eEFs). In particular, the extremely abundant translation factor EF1alpha would be expected to be an efficient binding competitor of vRNAP for charged tRNA^{Gln/Arg}. However, tRNA^{Gln/Arg} associated with vRNAP has been shown to be uncharged and is thus presumably disengaged from active translation and interactions with the aforementioned cellular factors⁸. This raises the question of how tRNA^{Gln/Arg} can be 'stolen' from the host for its activity in complete vRNAP assembly. As fully modified tRNAs are efficiently aminoacylated and likely sequestered by EF1alpha tRNA^{Gln} and tRNA^{Arg} lacking mcm5s2U may be less aminoacylated, as these hypomodified variants might avoid competition with EF1alpha and thus be preferentially utilized for vRNAP assembly. This might explain the observation that complete vRNAP contains exclusively uncharged tRNA^{Gln/Arg}. An alternative and not mutually exclusive scenario is that a local depletion of glutamine (and arginine) at viral replication sites may drive a shift from charged to uncharged tRNAs. Notably, such a situation may occur in the late phase of infection when complete vRNAP is formed for packaging and may provide a mechanism to link the metabolic status of the infected cell to the viral replication processes. In this regard, it is also noteworthy that glutamine is a crucial metabolite in viral energy metabolism^{28,29}.”

Hypomodified tRNA^{Gln} and tRNA^{Arg} are sequestered during vRNAP formation and later released during PIC assembly. Does this process have any impact on translation? In particular, codon optimality might be changed. A discussion from this perspective would be valuable.

We thank the reviewer for raising this important point. In our current study, we did not specifically investigate whether the sequestration and subsequent release of hypomodified tRNA^{Gln} and tRNA^{Arg} during vRNAP formation affects translation efficiency or codon optimality. However, we note that the amount of complete vRNAP-sequestered tRNA is very low as compared to the total pool of tRNA, making a direct impact on translation rather unlikely. Our findings lay a foundation for future research on the interplay between tRNA modifications, viral transcription machinery, and translation. Addressing how these tRNAs might be redirected toward the translation of viral transcripts, and whether codon optimization plays a role, is certainly a focus of our future studies.

The manuscript discusses three limitations to explain the selection of tRNA^{Gln} and tRNA^{Arg}. Are these criteria sufficient? Could other characteristics of tRNA play a role in this selection?

We believe that the three criteria that are discussed in the manuscript highlight the most important mechanism for tRNA selection. In fact, the analysis of mutant tRNA^{Gln} (U33G) shown in the new Extended Data Fig. 8 would argue that the induced fit mechanism enforcing mostly these three characteristics of the stem/loop is, apart from the modification status the major determinant for assembly activity.

However, we note that there is another spot in the elbow region where also a specific readout takes place. This involves the stabilization of an alternative loop conformation by the polymerase. Although these features may further improve the activity of tRNA^{Gln/Arg} in the assembly pathway, they are certainly not the main determinants.

We have thus refrained from mentioning these aspects in the manuscript in favor of a more stringent data presentation and discussion section.

The distorted structural changes in the ASL within the N-lobe of NPH-I are intriguing, but the description of Fig. 5F is insufficient. A more detailed explanation of the interactions between U35, A37, and the amino acid residues in the pocket, as well as the positioning of Mg²⁺ ions, is needed.

We have added in our revised version of the manuscript further depictions of the interaction details for U35, A37, and the magnesium ion in Fig. 5G.

The ASL model of tRNA in Fig. 5BC requires improvement. In this model, A37 and U33 are paired; however, in a free form of tRNA, the anticodon bases and positions 37 and 38 are stacked, while U33 typically forms a U-turn structure.

Thank you for this thoughtful comment. Indeed, the state shown emphasizes a possible A-U pairing, which does not represent the energetic minimum in solution. We have corrected our manual model according to your suggestions.

Reviewer #2 comment 6

The claim that mcm5s2U is low in uninfected cells and increases during infection is supported by the nucleoside analysis, as illustrated in Fig. 6CD.

Thank you for your comment. The data highlight how the global amount of mcm5s2U in total tRNA changes before and after infection. They also demonstrate that mcm5s2U in total tRNA was not entirely erased due to infection and confirm that our LC/MS protocol is reliable for detecting mcm5s2U modifications in tRNA.

However, there is no information about normalization of the peak intensity in the experiment.

We used the same input amount of total tRNA mixture for digestion and subsequent LC/MS analysis, as determined by absorbance at 260 nm. This is reflected in the UV peak intensities of the nucleosides after digestion. For normalization and quantitative comparison we have

Originally submitted

Revised

included Y-axis labels for the UV260 trace (see Fig. Below).

More importantly, mcm5s2U nucleoside levels can be influenced by the abundance of the four tRNA species carrying mcm5s2U. This data does not provide any information about tRNA modification status without quantifying the tRNA abundance.

We acknowledge this limitation. Changes in the abundance of the tRNA species are likely reflected in the slightly different ratios of the four main nucleosides. However, even under total

tRNA digestion conditions, i.e. when tRNA^{Gln} at low abundance is in total tRNA, we observed a detectable amount of mcm5s2U. This supports the reliability of the analysis conditions, demonstrating that the target modification can be detected if present in the tRNA sample. Nevertheless, under these same experimental conditions, mcm5s2U was not detected in tRNA extracted from the vRNAP complex (where the relative abundance of the target tRNA^{Gln} is significantly higher than in the total tRNA condition). This result suggested that the tRNA^{Gln} associated with vRNAP contains none or only trace amounts of the mcm5s2U modification.

To quantify tRNA modification rates, it would be necessary to isolate and purify tRNAGlu and tRNAArg and analyze the anticodon-containing fragments to measure the ratio of modified to unmodified fragments.

In this study, we did not aim to quantitatively compare modifications between infected and uninfected samples. In our revised manuscript we have clarified this point and toned down any statements implying such comparisons:

Page 10

Before

Despite a massive upregulation of the mcm⁵s²U modification of tRNAs upon infection, tRNA^{Gln}(UUG) carrying this modification in the anticodon would be incompatible with its function in assembly (Fig. 7D).

After

Despite a global upregulation of the mcm⁵s²U modification in total tRNAs upon infection, tRNA^{Gln}(UUG) carrying this modification in the anticodon would be incompatible with its function in assembly (Fig. 7D).

Additionally, intermediates such as s2U34 and mcm5U34, which precede mcm5s2U34, should also be verified. cm5s2U is a precursor to mcm5s2U but is known to be converted to ncm5s2U and is scarcely present in cells.

We also checked the other precursors of mcm5s2U by extracted ion chromatograms (EIC); however, we did not detect these intermediates in the tRNAGln samples we analyzed (see Fig. Below).

The detection of cm5s2U is likely due to hydrolysis of the terminal methyl ester during tRNA preparation.

That is correct. In our experiments using synthetic mcm5s2U standards, we observed hydrolysis of the methyl ester and/or desulfurization under the same treatment conditions as those used for tRNA digestion. This is why we presented the cm5s2U EIC to account for this potential artifact (see Fig. below).

The data in Fig. 1B cannot be interpreted as showing that tRNA binding promotes the association of D1/D12 with vRNAP. Rather, the data suggest that the binding of D1/D12 to vRNAP further stabilizes tRNA binding. A comparison of the binding efficiency of D1/D12 with the I3 and I1 intermediates is necessary.

The binding efficiency of D1/D12 to core vRNAP (I1) is very low and can hardly be detected by biochemical means (see Extended Data Fig. 1C). However, once the intermediate I3 (composed of core vRNAP, E11, NPH-I and tRNA) has been formed, efficient binding is enabled (Fig. 1C, please note that the figure labelling has been changed in the revised version). Therefore, even though D1/D12 does not directly bind tRNA^{Gln/Arg}, it required the prior action and presence of this assembly chaperone to generate the binding platform for D1/D12 on I3. This mechanism is described now in detail in the main text.

Minor points:

CE(D1/D12) should be written as D1/D12 in text, because this heterodimer is shown as D1 and D12 separately in many Figures.

We have adjusted the nomenclature as suggested.

If authors use previous structural data (structure I1, PIC, relaxed tRNA model, etc.), the PDB ID should be shown for these structures in text, Figure legend and Data availability sections.

We have added the PDB codes where appropriate. No experimental structure for eukaryotic tRNA^{Gln} is available, this is a manual model. We have added a comment clarifying this to the figure legend for Fig. 5B.

Typo in Fig3 legend lane 7; I5: complete vRNAP→I6:completevRNAP.

We have now eliminated the term I6 throughout the manuscript and now use the term complete vRNAP in the text and relevant figures.

In Figure 5, the legend of (F) is lacking.

We have now added the missing legend.

The number of methylation position should be shown in uppercase. (ex. m1A, m5C)

This has been corrected throughout the text and figures

The model of U34 in Fig7C is actually s2U. Please correct it.

We corrected the figure as suggested.

Extended Fig1: B, the arrow indicating NPH-1 degradation product is not shown. You should label the sample name of each lane.

As suggested, we have now included an arrow to indicate the NPH-1 degradation product and labeled the sample names for each lane to ensure clarity and accuracy. Please note that this is now the new Extended Data Fig. 1A.

You should mention and discuss the weak binding between D1/D12 and NPH-1/E11 in main text.

There are indeed contact surfaces between these three components as can be seen in the structure of complete vRNAP. This may cause background binding activity of these factors among each other as we see it in our in vitro binding assays. We have now mentioned this in the text on p. 4:

“However, a weak background binding activity can be attributed to existent invariable contact surfaces between these three components as documented in the structure of complete vRNAP.”

You should label the sample name of each gel in Figure.

As suggested, we have now labelled the different gradients accordingly.

Extended Fig2: B, I'm interested in the difference among Class 1 to 3. Please address it in figure legend or main text. Final resolution of 3.3A is not consistent with FSC curve data (Ex Fig2B). The letter '3.8A' to the left of the letter 'E' is probably a mistake.

We thank the referee for mentioning these mistakes. We have corrected the labels.

Class 1 and 2 could not be refined or classified further to refinable subclasses. It appears that they contain predominantly damaged or mispicked particles. We have added a corresponding label. For the same reason, we have not compared these two classes to the 'good' class 3.

Extended Fig4: B, The map of class 2 from 1st Classification is slightly strange. (The density at the top is distorted.)

We thank the referee for bringing up this point. In our hands, streaking artifacts in the background like observed here are relatively common with the Cryosparc refinement procedure. Apart from other reasons like thick ice, low S/N ratio, etc., refinement on a heterogeneous particle population may cause this. The latter is obviously the case here since further classification resolves the artifacts.

You should mention about class 1 from 2nd classification in main text. I'm also interested in remaining particles (about 240K particles) in 3rd classification. Is it only junk, or other conformations?

The three rounds of focused classification were used to sharpen the two major classes. Most eliminated particles belonged to only slightly different conformations regarding NPH-I, obviously representing a small, continuous movement. Given the large number of available particles, we could 'afford' to eliminate them in favor of a sharper appearance of the two classes. Some particles are actually junk. We now document this precisely in the figure.

Extended Fig5; You should clearly label '15' in Figure or Figure legends.

We agree and have labelled class 2 accordingly.

Extended Data Table1; You should clearly label 'Intermediate 5' to 8C8H column.

We have labelled this column accordingly.

You should write the number of ions in the same way in all column.
Particles (automatically selected) of Intermediate 3 and 4 are mistaken.
Number of movies of Intermediate 3 is mistaken.

This was corrected as suggested by the reviewer.

P46 lane4 Fig1C→ Fig1B

This was corrected as suggested by the reviewer

P46 lane11 the amounts of E11 (9 nmols) is correct?

The amount of E11 that was given was indeed incorrect. The amount of E11 that was used in the reconstitution experiment was 900 pmol. We corrected this in the relevant section.

P47 lane 7 I can't find total calf tRNA of Roche using this number.

Thank you for bringing this to our attention. The original batch of total calf tRNA used in our experiment was obtained from Roche, but it is no longer commercially available. A suitable alternative is calf tRNA from Sigma-Aldrich (catalog number R4752), which can be used in place of the Roche product without affecting the outcome of the experiments.

P47 You need to show the purification protocol for recombinant E11.

We appreciate the reviewer's interest in the experimental details of the recombinant E11 purification protocol. The procedures for expression, purification, and crystallization of vaccinia virus E11 can be found in a recent paper from our lab (Grimm *et al.* (2019)), which is cited in the manuscript.

P49 You should address the details of purification protocol of vRNAP samples in separate section. I can't find the details of the preparation of Intermediate 5 sample.

Thank you for highlighting the need for a clearer explanation of the purification protocol for vRNAP samples, including the preparation of the Intermediate 5 (I5). The general purification procedure is described in detail in a STAR protocol (Bartuli *et al.*, STAR protocols 3 (1), 101116) that is cited in the manuscript. No additional biochemical purification strategy was employed to isolate I5. As now mentioned in the manuscript and illustrated in the cryo-EM imaging processing scheme (Extended Data Fig. 5), I5 was identified as a subpopulation of particles on the cryo-EM grids used for the complete vRNAP complex. Thus, the I5 particles were present under the same purification conditions as the complete vRNAP and did not require a separate isolation procedure.

Reviewer # 3 (Remarks to the Author):

A previous landmark study (Cell, 2019) by Prof. Fischer and colleagues showed that the transcriptionally active "complete" vaccinia virus RNA polymerase complex comprised 15 viral polypeptides and uncharged host tRNA-Gln (and to lesser extent host tRNA-Arg). tRNA-Gln interacts with Rap94 (required for early gene transcription initiation), VETF (early promoter binding initiation factor) and NPH-I (a DEXH-box transcription termination factor). They proposed that tRNA is important for stability of the vRNAP complex.

Here, Bartuli *et al.* test their hypothesis that tRNA serves an assembly chaperone for

vRNAP. Starting with an 8-subunit minimal RNAP (lacking Rap94) and “core” RNAP (containing Rap94), they sought to reconstitute complete vRNAP by adding back purified components. They find that NPH-I and E11 suffice for recruitment of 3' 32P-labeled tRNA into the core RNAP complex, generating an assembly intermediate that was capable of recruiting the vaccinia capping enzyme.

The authors then proceed to solve cryo-EM structures (2.5 to 3.9 Å resolution) of assembly intermediates and complete vRNAPs (+VETF, lacking capping enzyme or –VETF, +capping enzyme) that provide additional insights into the interactions of the tRNA with protein components and protein-protein interactions.

Focusing on the structure of the vRNAP-associated tRNA-Gln, versus a model of the native tRNA fold, they find that the wobble base U34 remains flipped out whereas the rest of the anticodon stem-loop adopts a different fold that they term “strained,” entailing rearranged nucleobase hydrogen bonds. They make an argument that tRNA-Gln and tRNA-Arg are uniquely capable of adopting this strained anticodon stem-loops conformation.

A valuable component of the study is the inclusion of an LC–MS analysis of the modification status of the nucleosides of vRNAP-associated tRNA-Gln, which revealed that the RNAP-bound tRNA-Gln has the expected ribose and base methylations but lacks the mcm5s2U wobble modification. They affirm this by showing that the RNAP-bound tRNA-Gln resists anticodon incision by γ -toxin (which cleaves at mcm5s2U).

We thank the reviewer for his/her comments on our manuscript.

Finally, the authors document enrichment of tRNA-Gln/Arg in infectious vaccinia and Mpx virions. Albeit they do not demonstrate that the tRNAs are associated directly with the virion RNAP. tRNA-Gly is nearly as abundant in vaccinia virions as tRNA-Arg. What is tRNA-Gly doing the the vaccinia virus particle – and is it associated with virion RNAP? Is tRNA-Gly also enriched in Mpx virions? Inclusion of a probe for tRNA-Gly in Fig. 8C would be useful. It is relatively easy to purify RNAP from virion cores (without any tags), so perhaps the authors would consider doing this to confirm which tRNAs are bound to RNAP in virions.

We have currently no experimental data addressing the question whether the presence of high amounts of tRNA^{Gly} in vaccinia virions points to a specific function.

Interestingly, Northern blot experiments that we have performed with Mpx virions show a similarly strong enrichment of tRNA^{Gly}. We would be prepared to include this information into a revised Fig. 8C. However, biochemical experiments that we have done with vaccinia exclude that this tRNA associates with vRNAP within virions (and in infected cells) and thus is unlikely to contribute to viral transcription, which is the main topic of our study.

At present, we cannot exclude, however, that this tRNA serves other functions, for example as a structural component of the virion as observed for other viruses. Studies addressing this interesting aspect are currently ongoing.

We would love to do biochemical experiments with vRNAP of Mpox. Unfortunately, however, we are not allowed to do these experiments in our labs (we are at present only allowed to use inactivated DNase/proteinase-treated samples for our experiments). The experiment we show in Fig. 8C therefore provides the only, as we think strong, evidence supporting the hypothesis that the Mpox vRNAP also requires tRNA^{Gln/Arg} for assembly of a complete vRNAP and its incorporation into virions.

The Discussion provides a concise summary of key points. I think the speculation that local depletion of Gln and Arg amino acids at viral replication sites during the late stage of vaccinia infection prompts the development of an uncharged, wobble-unmodified tRNA-Gln/Arg pool for association with vRNAP is a stretch, especially given that translation of viral late mRNAs remains quite vigorous at late times when assembly of new virions occurs. (Moreover, the cell culture medium (DMEM) for vaccinia virus growth is typically supplemented with glutamine, so it is not clear that Gln is limiting during virus infection.) Perhaps the authors might temper their speculations at end of Discussion.

We agree with the reviewer that this is speculation, and we have tempered our discussion regarding this aspect. We have also extended the discussion speculating that the tRNA modification status helps to divert the tRNA from translation to vRNAP assembly.

Overall, this is an important study, rigorously performed, that sheds light on the assembly and structure of the poxvirus RNAP responsible for early gene transcription. The reliance of a specific tRNA and a strained tRNA conformation (as well as lack of a base modification) to allow for tRNA function as an assembly chaperone has broad implications for an expanding view of tRNA's biological functions.

We thank the reviewer for this positive statement and for his/her excellent suggestions for improving our work.

! " #
 \$! %& ' ! ! ' (& !) * ! ! ! (& ! ' ! ! ! ! ! ! +
 ,) "- & % ! ! ! & ! ' ! ! ! " +
 - ! + ! (% & (& ! ! ! ! ! ! ' / '
 ! ! " ! ' %d (' ! ! ! " /
 0 ! ! ! ! ! ! ! ! ! / ! &\$! # ! %
 ! ! ! ! ! 1 ! & ! ! ! ! ! ! ! ! ! ! ! # ! ! ! ! ! !
 & % # ' ! ! ! ! ! # ! ! ! ! ! - ! "
 & ! - + / # & & 2 - ! ! # ! %& ! % ! ! !
 1 ! & ! ! ! ! ! & ' & ! " !

552 6 \$ * \$55

! ! " #
 \$ %&
 \$
 * () ()))
 ! &# ! ! ! " 7 ' & ("\$ 8 ! ' ! % ! ! ! #
 ! ' & ! ! ! ! ! (& # ! (' 8 ! ' ! % ! ! ! #
 ! ' g
 ! ! ! &# ! & ! ! " 7 ' & # ! ! ! % % & / ! &
 ! ! ! 3 ! ! ! "- ' ! ! ! %d / ! & & (! !
 ! ! ! 7 ' ! ! ! & ! & ! ! ! # ! ! ! / ! & !
 !